# Stratifin as a novel diagnostic biomarker in serum for diffuse alveolar damage

Noriaki Arakawa[1], Atsuhito Ushiki[2], Mitsuhiro Abe[3], Shinichiro Matsuyama[1], Yoshinobu Saito[4], Takeru Kashiwada[4], Yasushi Horimasu[5], Akihiko Gemma[4], Koichiro Tatsumi[3], Noboru Hattori[6], Kenji Tsushima[7], Keiko Miyashita[1], Kosuke Saito [1], Ryosuke Nakamura[1], Takeshi Toyoda[8], Kumiko Ogawa [8], Motonobu Sato[9], Kazuhiko Takamatsu[9], Kazuhiko Mori [10], Takayoshi Nishiya[10], Takashi Izumi[11], Yasuo Ohno[11], Yoshiro Saito [1] ✉ & Masayuki Hanaoka[2]

Among the various histopathological patterns of drug-induced interstitial lung disease (DILD), diffuse alveolar damage (DAD) is associated with poor prognosis. However, there is no reliable biomarker for its accurate diagnosis. Here, we show stratifin/14-3-3σ (SFN) as a biomarker candidate found in a proteomic analysis. The study includes two independent cohorts (including totally 26 patients with DAD) and controls (total 432 samples). SFN is specifically elevated in DILD patients with DAD, and is superior to the known biomarkers, KL-6 and SP-D, in discrimination of DILD patients with DAD from patients with other DILD patterns or other lung diseases. SFN is also increased in serum from patients with idiopathic DAD, and in lung tissues and bronchoalveolar lavage fluid of patients with DAD. In vitro analysis using cultured lung epithelial cells suggests that extracellular release of SFN occurs via p53-dependent apoptosis. We conclude that serum SFN is a promising biomarker for DAD diagnosis.

Pharmaceutical products can cause lung injury as an adverse drug reaction; drug-induced interstitial lung disease (DILD) is a common example. In fact, studies have found that epidermal growth factor receptor tyrosine kinase inhibitors (EGFR-TKIs) such as gefitinib and erlotinib frequently cause DILD[1–3]. Although DILD is a problem in Europe and the US (with an incidence of 0.3% for gefitinib), the incidence rate of DILD is higher in Japan (2–3% for gefitinib)[4–7]. The high incidence rate in Japan is considered a serious issue not only in the clinical setting, but also in the context of global drug development.

There are several types of DILD, categorized based on histological patterns. The most serious type is diffuse alveolar damage (DAD). Of note, DAD is frequently observed in patients who have developed DILD due to the administration of tyrosine kinase inhibitors (TKIs), including EGFR-TKIs[8], and is also a common characteristic among the majority of patients with acute lung injury, such as cases of acute exacerbation (AE) of idiopathic pulmonary fibrosis (IPF) and acute respiratory distress syndrome (ARDS). Importantly, patients with DAD do not sufficiently respond to treatment, and their prognosis is poor; additionally, even if they recover from the disease, they continue to suffer from fibrosis. Therefore, for the diagnosis of DILD, early identification of patients with DAD is necessary in order to select an appropriate treatment and avoid severe outcomes or DILD-related death.

[1]Division of Medicinal Safety Science, National Institute of Health Sciences, Kanagawa 210-9501, Japan. [2]First Department of Internal Medicine, Shinshu University School of Medicine, Matsumoto 390-8621, Japan. [3]Department of Respirology, Graduate School of Medicine, Chiba University, Chiba 260-8670, Japan. [4]Department of Pulmonary Medicine and Oncology, Graduate School of Medicine, Nippon Medical School, Tokyo 113-8603, Japan. [5]Department of Respiratory Medicine, Hiroshima University Hospital, Hiroshima 734-8551, Japan. [6]Department of Molecular and Internal Medicine, Graduate School of Biomedical & Health Sciences, Hiroshima University, Hiroshima 734-8551, Japan. [7]Department of Pulmonary Medicine, School of Medicine, International University of Health and Welfare, Chiba 286-8686, Japan. [8]Division of Pathology, National Institute of Health Sciences, Kanagawa 210-9501, Japan. [9]Astellas Pharma Inc, Tsukuba 305-8585, Japan. [10]Daiichi Sankyo RD Novare Co., Ltd, Tokyo 134-8630, Japan. [11]Kihara Memorial Foundation, Yokohama 230-0045, Japan. ✉e-mail: yoshiro@nihs.go.jp

Biopsy for pathological evaluation is needed for the definitive diagnosis of DILD, but it is clinically and ethically difficult to obtain samples from patients with DAD because of their severe symptoms. Currently, high-resolution computed tomography (HRCT) chest scans are the most effective non-invasive method to diagnose the histological pattern of DILD[9]. However, it is difficult for general physicians other than respiratory specialists to correctly diagnose DAD by HRCT, because this method requires specialized knowledge and training. In addition, HRCT is highly expensive and poses a significant financial burden on health-care systems. Moreover, exposure to radiation during HRCT scanning is associated with an increase in the long-term risk of complications. Therefore, it is necessary to develop a biomarker that can be used specifically for the diagnosis of DAD.

Currently, the following biomarkers are used in the clinical detection of interstitial pneumonia: surfactant protein (SP)-A, SP-D, and Krebs von den Lungen-6 (KL-6); all of these are high-molecular-weight glycoproteins that are expressed by type II pneumocytes[10, 11]. KL-6 has been reported to be useful for the detection of EGFR-TKI-induced DILD in patients with non-small-cell lung carcinoma[12]. However, although these biomarkers can detect interstitial pneumonia in general, they are not used specifically for the diagnosis of DAD. Moreover, there is no correlation between these biomarkers and the severity of lung injury. In addition to DILD, several blood biomarkers have been proposed for the detection of acute lung injury, including acute interstitial pneumonia and ARDS. Such biomarkers include the molecular chaperone HSP47[13], the inflammatory cytokines IL-1β, IL-8, and IL-6; and the inflammatory mediators HMGB1 and LBP[14]. However, none of these biomarkers are currently used in the clinical setting.

In this study, we performed an aptamer-based proteome analysis using blood samples from DILD patients in order to identify a new diagnostic marker of DAD, and we validated the DAD-diagnostic performance of the thus-identified marker using two independent cohorts. Furthermore, to compare the clinical usefulness and features of the new identified biomarker with those of known biomarkers, we carried out exploratory analyses using the combined sample cohort for discriminating DAD type DILD from related lung diseases.

## Results

### Demographics of the participants

We performed a sample collection at the acute and recovery phases from consecutively recruited DILD-onset patients exhibiting the DAD pattern, the DAD pattern mixed with other dominant DILD patterns (DAD-mixed patterns), the organizing pneumonia (OP) pattern, the nonspecific interstitial pneumonia (NSIP) pattern, or the hypersensitivity pneumonitis (HP) pattern. To search for biomarkers specific to patients in the DAD group (the DAD and DAD-mixed patterns), we performed a proteomics study by SOMAscan assay in a "Discovery cohort" of patients with DILD who were enrolled in the early phase of this study, and validated the results using a "Validation cohort" consisting of additional enrolled DILD patients. Tolerant controls (who were administered similar drug(s) but exhibited no DILD onset) and various lung disease patients were also analyzed for biomarker evaluation.

The clinical characteristics of these cohorts are shown in Table 1 (for age, sex). In the Validation cohort, there was a significant difference in sex distribution between the DAD group and non-DAD group. Although significant differences were observed in age between healthy volunteers and the DAD group, there were no significant differences in age or the acute/recovery sample composition between the DAD and non-DAD groups in either the Discovery or Validation cohort. In addition, no obvious differences were found in the white blood cells test values, but significant differences were sometimes observed in the P/F ratio, SpO$_2$, and C-reactive protein (CRP) values in the non-DAD group, the DILD recovery group and Disease controls, when compared to those in the acute DAD group, indicating a severe and statistically

significant decline in respiratory function and inflammation status (Supplementary Table 1). The underlying diseases and the suspected causal drugs of DILD in the registered DILD patients were diverse (Supplementary Tables 2 and 3). Clinical information of individual was shown in Supplementary Data 1.

### Screening and validation of new biomarkers

First, we performed a SOMAscan assay-based proteomic analysis using plasma samples collected from subjects in the Discovery cohort (Supplementary Data 2). Using the quantitative data on 1310 proteins, we searched for proteins whose expression was particularly changed in the DAD group, i.e., proteins that were increased (fold change (FC) > 2) or decreased (FC < 0.5) in acute-phase patients with DAD versus the control group (healthy volunteers and patients in recovery). A total of 55 proteins met these criteria (listed in Supplementary Table 4). Next, we focused our analyses on the following 8 proteins, due to their high effect size ($g \geq 1.9$) in the group of patients with acute DAD: the upregulated macrophage-capping protein (CAPG), C-C motif chemokine 18 (PARC), stratifin/14-3-3σ (SFN), interleukin-1 receptor antagonist (IL-1Ra) and secreted phospholipase A2 (sPLA2), and the downregulated carbonic anhydrase 6 (CA6), kallistatin (KAL), and apolipoprotein AI (Apo-AI) (Supplementary Table 5). Among these candidates, SFN caught our attention because its levels were markedly increased in the DAD group (FC: 2.3-fold, $g = 1.9$), and did not change in patients with the OP and NSIP patterns (FC: 1.0 and 1.2, respectively; $g$ values = 0.1 and 0.6, respectively; Supplementary Table 5). Therefore, we next developed an in-house ELISA for SFN to validate the SOMAscan results. As shown in Supplementary Table 6, the results indicated that the analytical performance of SFN ELISA was sufficient to warrant further evaluation of this protein. In a parallel analysis using commercial kits, we measured the levels of sPLA2, PARC, IL-1Ra, Apo-AI, KAL, and KL-6 and SP-D, which are known to be representative biomarkers of interstitial pneumonia (Supplementary Data 3), to evaluate their diagnostic performance as DAD-specific biomarkers. Both the results of the assay by ELISA and the results using the commercial kits aligned with the SOMAscan data, except for IL-1Ra (data not shown; no replication of SOMAscan data by ELISA). As expected, the levels of SFN (Fig. 1) and all other candidates (Supplementary Fig. 1) were changed in acute DILD patients compared to healthy volunteers and returned to levels close to those in healthy volunteers in the recovery phases. Importantly, unlike the levels of KL-6 and SP-D, which were elevated not only in the DAD group but also in the non-DAD group (Fig. 1d, f), the SFN levels were significantly higher in the DAD group than the non-DAD group ($p < 0.0001$, Fig. 1b).

Next, we validated these findings by measuring these biomarker candidates in DILD patients and healthy volunteers in the Validation cohort. Importantly, the concentrations of each biomarker candidate in DILD patients in the Validation cohort were similar to those obtained in the Discovery cohort (Fig. 1 and Table 2 show the results for SFN, KL-6, and SP-D; Supplementary Fig. 1 and Supplementary Table 7 show results for theother candidates). Based on the receiver operating characteristic (ROC) curve analysis, the area under curve (AUC) values obtained with the Discovery cohort were reproduced in the Validation cohort (Fig. 1h and Supplementary Fig. 1m). The AUC values observed in the Validation cohort indicated that all the candidates, except for PARC and Apo-AI, had an extremely good performance (AUC > 0.98) for the discrimination between healthy volunteers and the DAD group. However, in terms of the determination of disease activity in patients with DAD (DAD group patients in the acute phase *versus* all DILD patients in the recovery phase), SFN (AUC [95% CI]: 0.93 [0.84–1.0]) was superior to KL-6 (0.65 [0.48–0.82], SP-D (0.84 [0.71–0.97]) (Fig. 1h), KAL (0.90 [0.81–1.0]) and other candidates (around 0.80) (Supplementary Fig. 1m). In addition, SFN (0.85 [0.73–0.97]) showed the highest DAD-diagnostic performance (discrimination between the DAD group and the non-DAD groups) among these candidates.

**Table 1 | Sample cohorts used in this study: numbers and characteristics**

| Group | | Pattern | | No. of cases (female) | Age (range) | No. of samples (acute/recovery) |
|---|---|---|---|---|---|---|
| Discovery cohort [95] | HV | | | 24 (12) | 61 (55–65)**, a, # | 24 |
| | DILD | **DAD group** | | **10 (2)** | **71 (56–86)** | **17 (10/7)** |
| | | | DAD§ | 6 (1) | 72 (61–86) | 11 (6/5) |
| | | | DAD-mixed† | 4 (1) | 70 (56–82) | 6 (4/2) |
| | | **non-DAD group** | | **30 (13)** | **61.5 (32–85)ns, #** | **54 (30/24)** |
| | | | OP | 13 (4) | 69 (32–85) | 22 (13/9) |
| | | | NSIP | 15 (6) | 63 (46–81) | 28 (15/13) |
| | | | Other‡ | 2 (1) | 61.5 (50–73) | 4 (2/2) |
| | | KW test | | ns | $p = 0.0018$** | ns |
| Validation cohort [120] | HV | | | 53 (33)ns, # | 34 (25–64)****, b, # | 53 |
| | DILD | **DAD group** | | **16 (3)** | **69 (54–84)** | **23 (16/7)** |
| | | | DAD§ | 11 (3) | 70.5 (60–80) | 13 (11/2) |
| | | | DAD-mixed† | 5 (0) | 67 (54–84) | 10 (5/5) |
| | | **non-DAD group** | | **28 (15)*, c, #** | **72.5 (52–79)ns, #** | **44 (28/16)** |
| | | | OP | 17 (11) | 72 (52–79) | 28 (17/11) |
| | | | NSIP | 7 (2) | 73 (63–77) | 10 (7/3) |
| | | | Other‡ | 4 (2) | 71.5 (56–75) | 6 (4/2) |
| | | KW test | | $p = 0.0093$** | $p < 0.0001$**** | ns |
| Tolerant controls [31] | Tolerant controls | | | 31 (12)ns, # | 69 (33–83) | 31 |
| Disease controls [186] | Lung cancer | | | 58 (17)ns, # | 72 (44–81) | 58 |
| | Infectious | | | 19 (3)ns, # | 75 (36–81) | 19 |
| | NTM | | | 14 (9)ns, # | 65.5 (51–81) | 14 |
| | IIPs | | | 43 (8)ns, # | 72 (41–83) | 43 |
| | CTD | | | 25 (16)ns, # | 67 (50–83) | 25 |
| | COPD | | | 15 (2)ns, # | 67 (51–80) | 15 |
| | BA | | | 12 (9)ns, # | 60 (43–87) | 12 |
| Combined DAD group [26] | | | | 26 (5) | 71 (54–84) | 40 (26/14) |
| | KW test | | | $p < 0.0001$**** | ns | ns |

Differences among the groups were first tested using the Kruskal–Wallis (KW) test. In the Discovery and Validation cohorts, healthy volunteers, the DAD group and the non-DAD group were compared (comparison a). The tolerant and disease controls were compared with the combined data of the DAD group in the Discovery and Validation cohorts (comparison b). When significant differences were observed in the KW tests, then Dunn's test was performed comparing the DAD group with healthy volunteers and the non-DAD group (comparison a) or comparing the combined DAD group with disease controls (comparison b).

Data of the DAD and non-DAD groups are shown in bold.

ns not significant, HV healthy volunteer, IIPs idiopathic interstitial pneumonias, CTD lung disease associated with connective tissue disease, COPD chronic obstructive pulmonary disease, NTM nontuberculous mycobacteria, BA bronchial asthma, infection bacterial and mycotic pneumonia, OP organizing pneumonia, NSIP nonspecific interstitial pneumonia.

#Differences in the DAD group or combined DAD group were assessed by the Dunn's test. *$p < 0.05$; *$p < 0.01$; ***$p < 0.001$; ****$p < 0.0001$.

§DAD: patients diagnosed with a DAD-only pattern or DAD-dominant pattern (examples of diagnosis: DAD > NSIP, DAD > OP).

†DAD-mixed: patients diagnosed with DILD subtypes other than a DAD-dominant pattern, but in whom co-presence of the DAD pattern was observed by HRCT (examples of diagnosis: OP > DAD, HP > DAD, DAD = HP).

‡Other: cases including eosinophilic pneumonia (EP) and hypersensitivity pneumonitis (HP).

a$p = 0.001$ compared with the DAD group (Discovery cohort).

b$p < 0.0001$ compared with the DAD group (Validation cohort).

c$p = 0.030$ compared with the DAD group (Validation cohort).

## Comparison of disease specificity in the Combined sample cohort

To evaluate these candidates by comparing disease specificities, we performed an exploratory analysis using the Combined cohort (with combined data from the Discovery and Validation cohorts and controls) (Supplementary Data 3). Figure 2 shows the distribution of the levels of SFN, SP-D, and KL-6 in patients with various lung diseases. The corresponding values are shown in Supplementary Table 8. The levels of SP-D and KL-6 were elevated not only in the acute phase in all types of DILD patients but also in patients with idiopathic interstitial pneumonias (IIPs) and lung diseases associated with connective tissue diseases (CTDs) (Fig. 2a, b). In contrast, the SFN levels were markedly elevated in a specific fashion in patients with the DAD and DAD-mixed patterns when compared those in the DILD patients with the non-DAD pattern, the tolerant controls or the seven disease controls ($p < 0.0001$), although infrequent increases of SFN levels were

observed in the non-DAD patients of the DILD group, patients with lung carcinoma and IIPs (Fig. 2c). A high level of SFN was not observed in patients with CTDs or infectious lung pneumonia, which are important diseases to discriminate from DILD ($p < 0.0001$, Fig. 2c). Meanwhile, the levels of sPLA2, PARC and KAL were changed in patients with infectious pneumonia, and the level of Apo-AI was dispersed within each patient group (Supplementary Fig. 2; the corresponding values are shown in supplementary Table 8 and Supplementary Data 3). Thus, for all of these protein candidates, the serum SFN level showed relatively specific association with DAD.

## Comparison of the biomarker performance in the Combined cohort

Next, following on the above-described results, we performed a ROC analysis to precisely compare the biomarker performance of SFN to that of the known biomarkers in the Combined cohort (Fig. 3). In terms

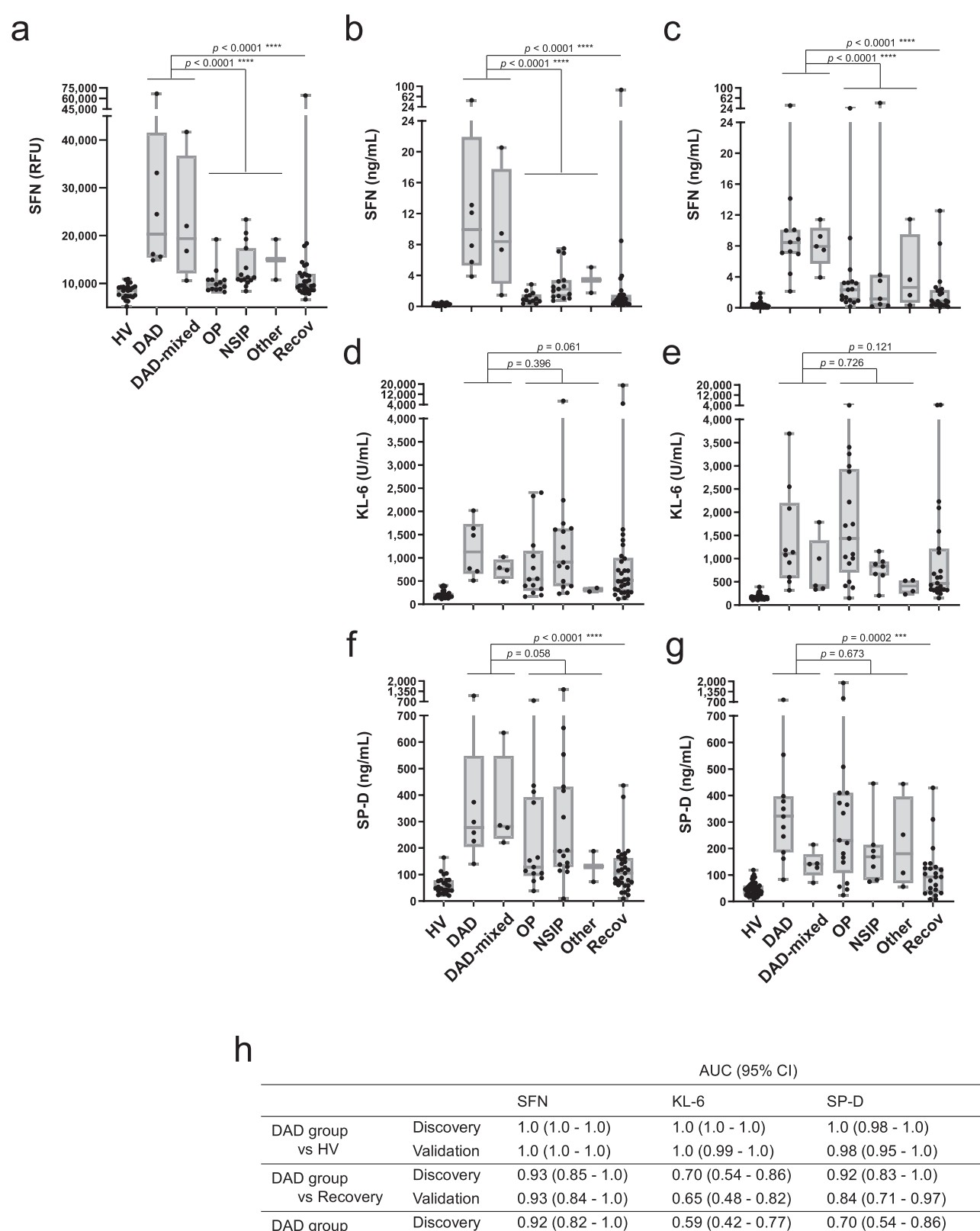

| | | AUC (95% CI) | | |
|---|---|---|---|---|
| | | SFN | KL-6 | SP-D |
| DAD group vs HV | Discovery | 1.0 (1.0 - 1.0) | 1.0 (1.0 - 1.0) | 1.0 (0.98 - 1.0) |
| | Validation | 1.0 (1.0 - 1.0) | 1.0 (0.99 - 1.0) | 0.98 (0.95 - 1.0) |
| DAD group vs Recovery | Discovery | 0.93 (0.85 - 1.0) | 0.70 (0.54 - 0.86) | 0.92 (0.83 - 1.0) |
| | Validation | 0.93 (0.84 - 1.0) | 0.65 (0.48 - 0.82) | 0.84 (0.71 - 0.97) |
| DAD group vs non-DAD | Discovery | 0.92 (0.82 - 1.0) | 0.59 (0.42 - 0.77) | 0.70 (0.54 - 0.86) |
| | Validation | 0.85 (0.73 - 0.97) | 0.53 (0.35 - 0.71) | 0.54 (0.37 - 0.71) |

of the determination of disease activity (acute *versus* recovery patients) in the DAD group, the performance of SFN (AUC [95% CI] 0.93 [0.88–0.99]) was higher than the performances of SP-D (0.86 [0.78–0.95]) and KL-6 (0.68 [0.56–0.80]) (Fig. 3a). Moreover, in terms of the diagnostic performance for discriminating the DAD group from the non-DAD group, SFN (0.90 [0.83–0.97]) was superior to SP-D and

KL-6 (AUC ≤ 0.61, Fig. 3b). In addition, the discriminating performance of SFN (0.92 [0.83–1.0]) between the DAD group and tolerant control group (no DILD onset) was almost equivalent to that of SP-D (0.95 [0.89–1.0]), Fig. 3c).

Importantly, the differential diagnosis of DILD depends on the exclusion of infectious pneumonia. Therefore, we also determined the

**Fig. 1 | Blood levels of SFN and known biomarkers for the detection of acute DILD.** SOMAscan signals of SFN (**a**) and serum levels of SFN (**b**, **c**), KL-6 (**d**, **e**) and SP-D (**f**, **g**) measured by ELISA (**b**, **c**) or clinical chemistry kits (**d**–**g**) in healthy volunteers (HV) and DILD patients. **a**, **b**, **d**, **f** Discovery cohort: HV (*n* = 24), DAD (*n* = 6), DAD-mixed (*n* = 4), OP (*n* = 13), NSIP (*n* = 15), other (*n* = 2), Recov (*n* = 31). **c**, **e**, **g** Validation cohort: HV (*n* = 53), DAD (*n* = 11), DAD-mixed (*n* = 5), OP (*n* = 17), NSIP (*n* = 7), other (*n* = 4), Recov (*n* = 23). The boxes indicate interquartile ranges (75% and 25%) and medians; whiskers show min and max values. Differences between the two groups were compared by a two-tailed Mann–Whitney *U*-test (unadjusted): **$p < 0.01$; ***$p < 0.001$; ****$p < 0.0001$. DAD diffuse alveolar damage, OP organizing pneumonia, NSIP nonspecific interstitial pneumonia, Recov all recovery patients. **h** The area under the curve (AUC) was derived from receiver operating characteristic (ROC) curves, and the 95% confidence intervals (95% CIs) were determined for the biomarkers in comparative analyses. Corresponding median (range) values for each group are shown in Table 2. Source data are provided as a Source data file.

**Table 2 | Comparison of the serum levels of SFN, KL-6, and SP-D in the Discovery and Validation cohorts**

| | | Concentration median (range), [*n*] | | |
|---|---|---|---|---|
| | | **SFN** | **KL-6** | **SP-D** |
| Measurement | Min. Dilution | 2-Fold | 51-Fold | 11-Fold |
| | LLoQ (unit) | 0.1 (ng/mL) | 10.0 (U/mL) | 17.2 (ng/mL) |
| HV | Discovery | 0.3 (0.1–0.6) | 198.6 (114.2–408.0) | 51.0 (20.2–164.2) |
| | Validation | 0.2 (0.1–1.9) | 162.7 (105.1–391.7) | 44.7 (8.6–118.4) |
| DILD acute | DAD | Discovery | 10.0 (3.9–48.3) | 1128 (512.0–2018) | 278.1 (140.1–1072) |
| | | Validation | 8.5 (2.1–34.4) | 1136 (316.2–5366) | 320.9 (82.7–828.4) |
| | DAD-mixed | Discovery | 8.4 (1.5–20.5) | 762.2 (482.5–1019) | 281.6 (220.7–635.1) |
| | | Validation | 7.9 (3.9–11.4) | 417.7 (334.1–1784) | 141.3 (70.5–214.4) |
| | OP | Discovery | 0.9 (0.3–2.8) | 538.6 (164.2–2408) | 128.1 (38.2–765.4) |
| | | Validation | 2.3 (0.1–24.1) | 1436 (150.5–4217) | 231.2 (24.6–1908) |
| | NSIP | Discovery | 2.3 (0.7–7.5) | 904.2 (229.0–6826) | 187.7 (8.6–1463) |
| | | Validation | 1.2 (0.1–43.9) | 831.3 (203.0–1156) | 168.4 (74.6–445.0) |
| | Other | Discovery | 3.4 (1.8–5.1) | 311.6 (273.4–349.9) | 130.6 (72.4–188.7) |
| | | Validation | 2.6 (0.3–11.5) | 408.8 (228.5–532.4) | 180.4 (56.2–443.8) |
| DILD recovery | All | Discovery | 0.8 (0.1–87.0) | 536.0 (112.7–18959) | 105.4 (8.6–436.7) |
| | | Validation | 0.8 (0.1–12.5) | 471.2 (149.4–4474) | 93.1 (8.6–428.8) |

The matrix used, the minimum dilution factors, the lower limits of quantification (LLoQ), and the concentration of each biomarker in healthy volunteers and DILD patients (median and range), as well as the number of samples measured (*n*) in the Discovery and Validation cohorts are indicated.

*DAD* diffuse alveolar damage, *OP* organizing pneumonia, *NSIP* nonspecific interstitial pneumonia.

diagnostic performance in this context. Again, SFN was the biomarker showing the strongest performance for discriminating between the acute phase of the DAD group and patients with fungal/bacterial pneumonia (0.98 [0.95–1.0]); the performance of SFN was superior to those of SP-D (0.87 [0.76–0.98]) and KL-6 (0.86 [0.74–0.98]) (Fig. 3d).

Next, we deepened our analysis and focused on the positivity rates; Table 3 shows a comparison of the positivity rates obtained with SFN, SP-D, and KL-6 in various lung diseases. For KL-6 and SP-D, the normal reference levels used in the clinical settings were set as cut-off values (500 U/mL and 110 ng/mL, respectively). For SFN, we applied the Youden Index-based cutoff value of 3.6 ng/mL for discriminating between the DAD group and the tolerant control group. Note that although the Combined cohort data (Fig. 3c) were used to calculate the cutoff value, similar values (in the range of 3.6–3.7 ng/mL) were obtained even when calculated separately for the Discovery and Validation cohorts, indicating the robustness of the cutoff value. Remarkably, none of the healthy volunteers tested positive for SFN, while a high SFN-positivity rate (92%) was observed in the acute-phase patients of the DAD group; meanwhile, the SFN-positivity rates were low in patients with the non-DAD pattern of DILD (10–33%), suggesting that serum SFN-based diagnosis of DAD is possible by setting an appropriate cut-off value for SFN. In addition, the positivity rates of SFN in the DILD patients in recovery, tolerant control patients (mostly lung cancer patients), and lung cancer patients (9%, 10%, and 19%, respectively) were lower than those for KL-6 (50%, 29%, 29%, respectively) and SP-D (44%, 29%, and 26%, respectively). Finally, in patients with infectious pneumonia (*n* = 19), while four and six patients were positive for KL-6 and SP-D, respectively, only one patient (pneumocystis pneumonia) was positive for SFN. Thus, SFN was found to have a good performance as a DAD-specific biomarker.

## SFN levels were not associated with age, sex, the underlying diseases, or causative medications

There were no correlations between the serum SFN levels and the age of the DILD patients (Supplementary Fig 3a). However, borderline differences (*p* = 0.032) in the SFN levels were found between males and females in the non-DILD group, but not between males and females in healthy volunteers, the DAD-group or the tolerant controls (Supplementary Fig 3b).

As previously mentioned, the DILD patients enrolled in this study had various underlying diseases and had been administered several drugs suspected to be the cause of their DILD. The underlying diseases of the DILD patients were classified into three disease groups: lung cancer, other tissue cancer (including pancreatic, esophageal, and breast cancers), and non-cancerous diseases (including heart failure and rheumatoid arthritis) (Supplementary Table 2 and Supplementary Data 1). The suspected drugs were categorized into the following five types in accordance with their mechanism of action (Supplementary Table 3 and Supplementary Data 1): DNA-damaging agents (DDAs, including platinum drugs, gemcitabine, irinotecan, bleomycin, and 5-fluorouracil); taxanes (paclitaxel and docetaxel); tyrosine kinase inhibitors (including EGFR-TKIs such as erlotinib and osimertinib, as well as VEGF inhibitors such as axitinib and bevacizumab); immune checkpoint inhibitors (ICIs, including nivolumab, and pembrolizumab); and other drugs (mTOR inhibitors and non-oncology drugs such as antibiotics, anti-rheumatoid drugs, antiarrhythmic agents, and Chinese herbal medicines). In the DAD group, there were no significant differences in the levels of SFN among patients with lung cancer, other tissue cancers, and non-cancerous diseases, and no significant differences among patients treated with the different classes of suspected drugs (Supplementary Fig. 4a and d); in fact, the SFN-positivity rates (>3.6 ng/mL)

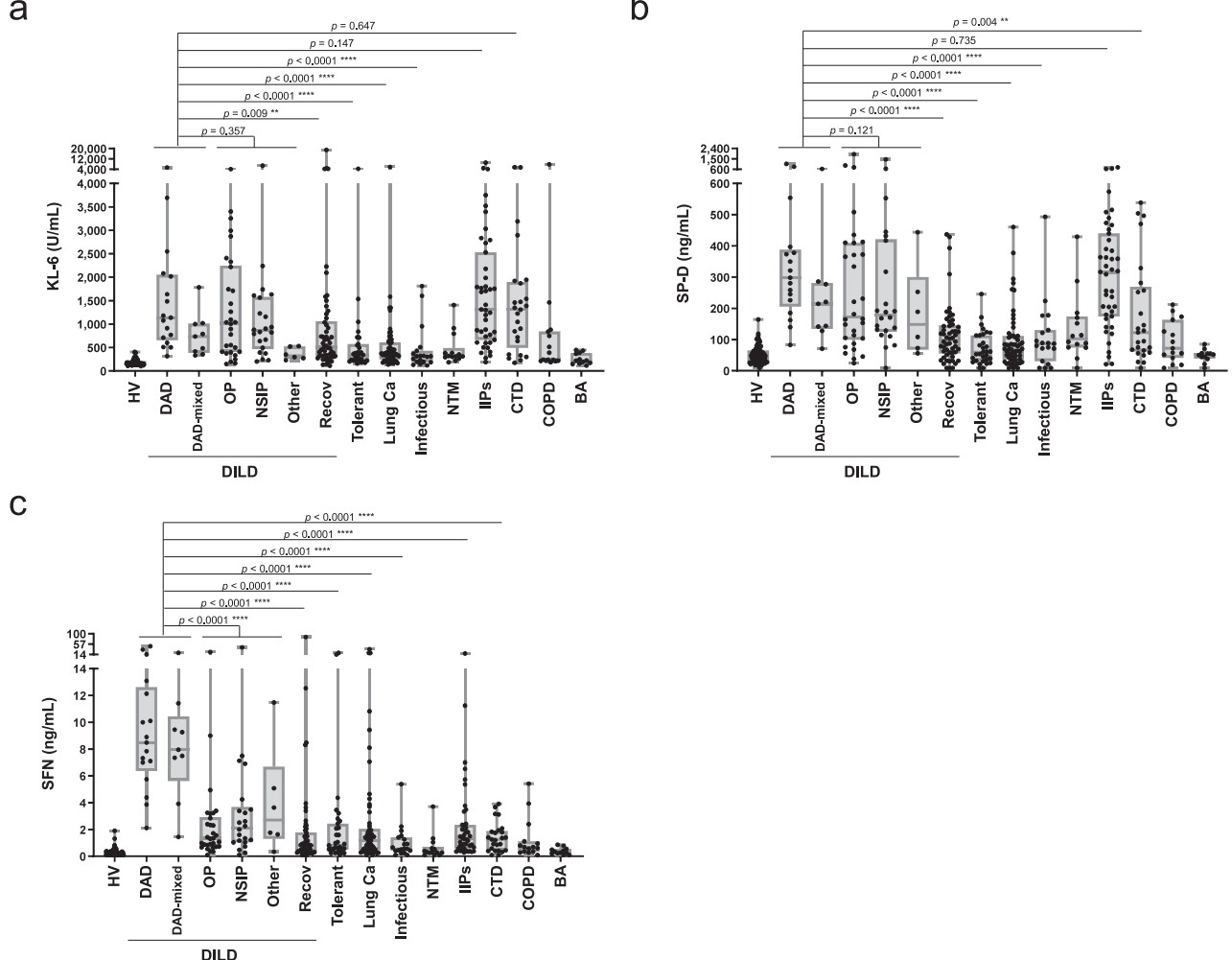

**Fig. 2 | Distribution of SFN and known biomarkers in healthy volunteers and patients with various lung diseases.** Results for the Combined cohort are shown. Serum levels of KL-6 (**a**), SP-D (**b**), and SFN (**c**) were measured by ELISA. Numbers of samples: HV ($n = 77$), DAD ($n = 17$), DAD-mixed ($n = 9$), OP ($n = 30$), NSIP ($n = 22$), Other ($n = 6$), Recov ($n = 54$), Tolerant ($n = 31$), Lung Ca ($n = 58$), Infectious ($n = 19$), NTM ($n = 14$), IIPs ($n = 43$), CTD ($n = 25$), COPD ($n = 15$) and BA ($n = 12$). The boxes indicate interquartile ranges (75% and 25%) and medians; whiskers show min and max values. Differences between the two groups were compared by a two-tailed Mann–Whitney $U$-test (unadjusted): **$p < 0.01$;

***$p < 0.001$; ****$p < 0.0001$. HV healthy volunteers, DAD diffuse alveolar damage, OP organizing pneumonia, NSIP nonspecific interstitial pneumonia, Recov all DILD patients in recovery, Tolerant tolerant control, Lung Ca lung cancer, IIPs idiopathic interstitial pneumonias, CTD lung disease associated with connective tissue disease, COPD chronic obstructive pulmonary disease, NTM nontuberculous mycobacteria, BA bronchial asthma, Infectious bacterial and mycotic pneumonia. Corresponding median (range) values for each group are shown in Supplementary Table 8. Source data are provided as a Source data file.

were almost equivalent among the groups (Supplementary Tables 10 and 11). In addition, in the non-DAD group and tolerant control patients, no clear relationships were observed between the levels of SFN and underlying diseases or causative drugs (Supplementary Fig. 4b, c, e and f). Collectively, these findings suggest that the increase in the serum levels of SFN in patients with DAD was not affected by the patients' age, underlying diseases, or the administration of any drugs. Although the SFN levels were significantly different between males and females in the non-DILD group, the SFN values in both sexes were mostly overlapped, and it appeared to have no clear impact on the significant differences in the SFN levels between the DAD and non-DAD groups.

**SFN levels were not related with DILD patient's outcomes**
Patients with DILD with the DAD pattern often develop severe symptoms and die. In this study, 27% (7/26) of the patients with DAD died of DILD; no DILD-related deaths were observed in the non-DAD group (Supplementary Data 1). No significant differences in SFN values were found between the patients who recovered from DAD-type DILD and those who did not (Supplementary Fig. 5a). In

addition, in the recovered patients in the DAD and non-DAD groups, there was no correlation between SFN values and the number of days required for DILD treatment (Supplementary Fig. 5b). These results indicate that the patients' SFN values did not necessarily predict patient outcomes.

**Changes in the biomarkers in AE-IIPs**
Among the 43 patients with IIPs analyzed (Fig. 2), AE-IIPs were observed in six patients (five with AE-IPF and one with acute interstitial pneumonia, Supplementary Data 1). In general, DAD is also a histopathologic hallmark of AE-IIPs[15]. Figure 4 shows the distribution of the blood levels of SFN and the two known biomarkers according to the categorization of patients with IIPs. While there was a tendency for the levels of KL-6 and SP-D to be higher in the patients with AE-IIPs, the levels of these biomarkers were also frequently higher (SP-D > 110 ng/mL, KL-6 > 500 U/mL) in patients with IPF and NSIP (Fig. 4b, c). In contrast, the levels of SFN were significantly higher in patients with AE-IIPs compared to patients with the other disease types, who were mostly negative for SFN (<3.6 ng/mL, Fig. 4a). Therefore, our data

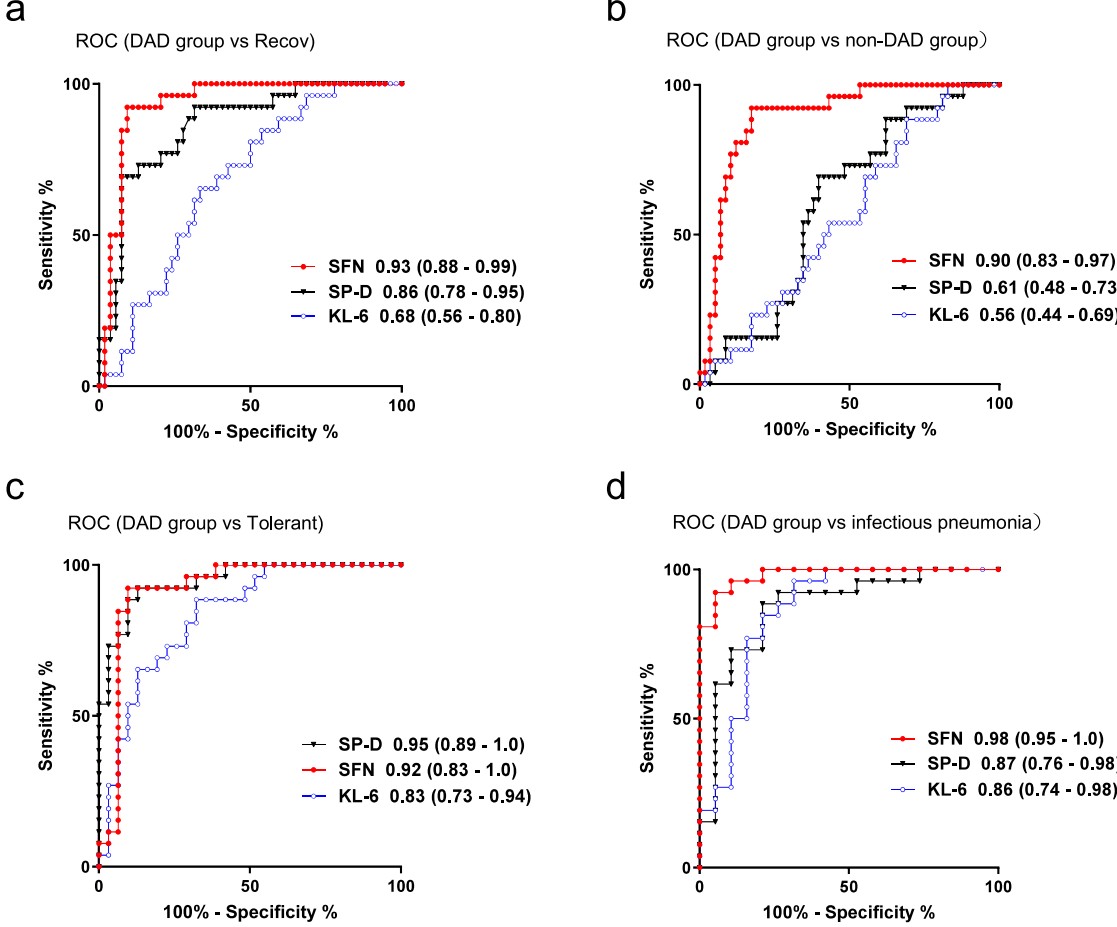

**Fig. 3 | ROC curves for SFN and known biomarkers.** Data of the Combined cohort are shown. ROC curves for SFN as well as the existing markers (SP-D and KL-6) were drawn to discriminate the acute phase of DAD patients from **a** all patients in recovery (discrimination of the disease activity), **b** acute-phase patients with other DILD types (DAD diagnostic performance), **c** the DILD-tolerant control patients (DAD onset), and **d** patients with infectious lung disease. Values in parentheses indicate the area under the curve (AUC) values. Source data are provided as a Source data file.

suggest that SFN is strongly associated with DAD, and may also be a useful marker for AE-IIPs.

## Origin of SFN

To clarify whether the increase of serum SFN levels in patients with DAD was a result of the release of SFN from the damaged lung tissues, we investigated the expression of SFN in lung tissues and bronchoalveolar lavage fluid (BALF) from patients with DAD. We immunohistochemically analyzed 19 lung tissue specimens, including 9 DAD autopsy specimens (from patients with DILD and with AE-IIPs such as ARDS) and 10 control tissues (five autopsy and five surgical specimens from patients with lung cancer). Among the nine DAD cases, only one was negative (−) for SFN expression, two had occasional slightly positive cells (±), two had some positive lesions (+), and four cases had several positive lesions (++). In contrast, among 10 control specimens, six were −, two were ±, one was + and 1 was ++. Cytoplasmic staining was observed mostly in the basal cells of the bronchioles (Fig. 5a-1) and some alveolar type II cells (Fig. 5a-2). Positive staining was observed in the cells that tended to show type II pneumocyte hyperplasia or squamous cell metaplasia, the pathological features of advanced DAD. The strength of expression was associated with the grade of lesions observed. In contrast, SFN expression was not observed in the cells of alveolitis with bleeding in the lung cancer case (Fig. 5a-3) or in the normal-looking alveolar cells (Fig. 5a-4) of controls. In the non-tumor specimens from the lung cancer patients, a few positive cells were observed in the basal cells of the bronchioles where focal interstitial inflammation was observed.

In addition, significant correlation was observed between the levels of SFN in serum and BALF, which were taken from patients with DILD or IIPs ($n = 14$, Spearman's rank correlation coefficient [$r_s$] = 0.670, $p = 0.0053$) (Fig. 5b). Moreover, there was also a significant correlation between the serum levels of SFN and the PaO$_2$/FiO$_2$ ratio, an indicator of blood oxygenation, in DILD patients ($n = 40$, $r_s = -0.439$, $p = 0.0023$) (Fig. 5c). These results suggest that SFN may be upregulated in damaged cells at the alveoli and bronchioles, and then released into the blood. This finding is supported by the significant correlations observed between the serum SFN levels and the SpO$_2$/FiO$_2$ ratio ($r_s = -0.434$, $p < 0.0001$), as well as between the serum levels of SFN and the serum levels of CRP ($r_s = 0.520$, $p < 0.0001$) and lactate dehydrogenase (LDH) ($r_s = 0.437$, $p < 0.0001$), which are indicators of inflammation and tissue injury, respectively (Supplementary Fig. 6).

## Extracellular release of SFN in vitro

To elucidate whether lung epithelial cells have the potential to release SFN, the expression of SFN was analyzed in vitro using the A549 cell line (derived from type II pneumocytes with an intact p53 gene) and a primary cultured small airway epithelial cells (SMECs). Because SFN is known to be a transcription target of p53[16], we first examined the expression of SFN in A549 cells treated with JNJ26854165 and Nutlin-3, which are small molecular compounds that activate p53 by inhibiting the interaction between p53 and MDM2. We also examined SFN expression in A549 cells treated with bleomycin and H$_2$O$_2$, which have also been demonstrated to induce activation of the p53-dependent

**Table 3 | Comparison of the positivity rates for SFN, KL-6, and SP-D in patients with DILD and three types of controls**

| Patient group | | %. Positivity rate (N, positive / N, analyzed) | | |
|---|---|---|---|---|
| | | SFN | KL-6 | SP-D |
| | Positive | >3.6 ng/mL | >500 U/mL | >110 ng/mL |
| HV | | 0 (0/77) | 0 (0/77) | 4 (3/77) |
| DILD acute | **DAD group** | **92 (24/26)** | **81 (21/26)** | **92 (24/26)** |
| | DAD | 94 (16/17) | 94 (16/17) | 94 (16/17) |
| | DAD-mixed | 89 (8/9) | 56 (5/9) | 89 (8/9) |
| | **non-DAD group** | **17 (10/58)** | **67 (39/58)** | **74 (43/58)** |
| | OP | 10 (3/30) | 70 (21/30) | 70 (21/30) |
| | NSIP | 23 (5/22) | 73 (16/22) | 86 (19/22) |
| | Other | 33 (2/6) | 33 (2/6) | 50 (3/6) |
| DILD recovery | All | 9 (5/54) | 50 (27/54) | 44 (24/54) |
| Tolerant control | | 10 (3/31) | 29 (9/31) | 29 (9/31) |
| Lung cancer | | 19 (11/58) | 29 (17/58) | 26 (15/58) |
| Infectious | | 5 (1/19) | 21 (4/19) | 32 (6/19) |
| NTM | | 7 (1/14) | 21 (3/14) | 50 (7/14) |
| IIPs | | 16 (7/43) | 86 (37/43) | 88 (38/43) |
| CTD | | 8 (2/25) | 76 (19/25) | 52 (13/25) |
| COPD | | 13 (2/15) | 40 (6/15) | 40 (6/15) |
| BA | | 0 (0/12) | 0 (0/12) | 0 (0/12) |

The reference values used in the clinical settings (500 U/mL and 110 ng/mL) were used as the cut-off for KL-6 and SP-D. The Youden's index-based cutoff of 3.6 ng/mL for SFN was obtained by comparing SNF levels in the DAD group and the tolerant control group.
Data of the DAD and non-DAD groups are shown in bold.
*DAD* diffuse alveolar damage, *OP* organizing pneumonia, *NSIP* nonspecific interstitial pneumonia, *IIPs* idiopathic interstitial pneumonias, *CTD* lung disease associated with connective tissue disease, *COPD* chronic obstructive pulmonary disease, *NTM* nontuberculous mycobacteria, *BA* bronchial asthma, *infectious* bacterial and mycotic pneumonia.

pathway, resulting in apoptosis or senescence in lung cells, including those of the A549 cell line[17–20]. Importantly, when A549 cells were treated with various concentrations of JNJ26854165 for 48 h, we observed a dose-dependent increase in the SFN (and LDH) levels released to extracellular media as well as a decrease in cell proliferation (Fig. 6a). Furthermore, the SFN upregulation and release induced by JNJ26854165 (10 μM) were observed in a time-dependent manner along with the accumulation of the transcription factor p53 (Fig. 6b, c). The elevations of p53 (and its phosphorylated forms at Ser15 and Ser392) were more pronounced from 24 h after exposure to the agent both in the cytoplasm and nucleus (Fig. 6c). On the other hand, the extracellular SFN levels were markedly increased along with the extracellular LDH levels after 48 h (Fig. 6b). Figure 6d shows the extracellular and intracellular protein levels, and mRNA levels of *SFN* in A549 cells at 48 h after exposure to various agents. The SFN expression was upregulated at both the intracellular protein and the mRNA level by all agents, but the level of extracellularly released SFN differed among the agents used. The SFN release was most pronounced after the JNJ26854165 (10 μM)-treatment, followed by the $H_2O_2$ (100 μM)-treatment, and was only slight after the bleomycin and Nutlin-3-treatments in A549 cells. In these cells, the levels of total p53 and/or its phosphorylated forms in the nucleus (Fig. 6e, under panel) as well as in the whole cell lysates (Fig. 6e, upper panel) were increased by all four agents, but these increases were not correlated with the levels of extracellularly released SFN, suggesting that the differences among the agents cannot be explained by p53 alone. On the other hand, we found that the increased levels of p21 were markedly higher in the bleomycin- and Nutlin-3-treated cells compared to the JNJ26854165- or $H_2O_2$-treated cells. This finding suggested that the previously reported anti-apoptotic effect of p21[21] had a greater impact on these slight SFN-releasing cells. Thus the extracellular release of SFN may depend not

only on the p53 activation, but also on the induction of apoptosis. In fact, the levels of extracellularly released keratin18 (K18) were shown to be elevated markedly, with the caspase-cleaved form of K18 (ccK18), an apoptosis marker, in the JNJ26854165- and $H_2O_2$-treated A549 cells; this also strongly suggested induction of apoptosis (Fig. 6f). Moreover, siRNA-mediated p53 suppression (Fig. 7a, b) or caspase inhibitor Z-VAD-FMK (Fig. 7c, d) attenuated the extracellular release of SFN (and LDH), also suggesting that p53-dependent apoptosis was involved in the SFN release. Lastly, we analyzed primary cultured SAECs to replicate the finding obtained using bleomycin, a more pathophysiologically related drug (commonly used as an ILD-inducer). The upregulation of SFN expression/release with p53 activation and its attenuation by Z-VAD-FMK were observed even in the SAECs (Fig. 7e, f). Taken together, these results suggest that the intracellular level of SFN may be increased in lung epithelial cells in a p53-dependent manner, and immediately released extracellularly upon induction of apoptosis.

## Discussion

This study sought to identify a blood-based biomarker of DILD and evaluated the usefulness of SFN as a DAD marker. In fact, until now, the relationship between SFN and DAD-type interstitial pneumonia has been unclear, and, to the best of our knowledge, no study has reported the detailed behavior of SFN in the blood. In a validation study using our established SFN ELISA, we observed that the serum levels of SFN were specifically increased in patients with DAD. Thus, our findings on SFN were reproduced in samples from an independent cohort. Moreover, the biomarker performance of SFN for discriminating DAD was superior to those of known biomarkers such as KL-6 and SP-D, which are currently used in clinical settings. In addition, our data suggest that SFN may be useful not only for the diagnosis of drug-induced DAD but also for the diagnosis of idiopathic DAD. We also showed that the levels of SFN were increased not only in the serum, but also in autopsy specimens and the BALF from patients with DAD. Serum SFN levels were significantly correlated with respiratory parameters. In addition, up-regulation and extracellular release of SFN were observed in response to the activation of p53 followed by apoptosis in pulmonary epithelial-derived cells. These findings suggest that serum SFN levels might reflect the degree of lung injury in patients with DAD; thus, this biomarker would be useful for the diagnosis of DILD patients with DAD as well as for the monitoring of their response to treatment.

SFN (14-3-3σ) belongs to the 14-3-3 family, together with six other isoforms, namely β, γ, ε, ζ, η, and τ. This family is a group of highly evolutionarily conserved proteins with a molecular mass of 25–30 kDa, expressed in all eukaryotes and involved in the modulation of various cellular processes via the binding to phosphorylated proteins[22]. While most 14-3-3 isoforms are expressed in various normal tissues in a ubiquitous manner, SFN is expressed specifically in stratified epithelia[23]. In fact, immunohistochemistry staining images of SFN included in the Human Protein Atlas database (https://www.proteinatlas.org) show that SFN is expressed permanently and abundantly in the squamous epithelium of the skin and esophageal tissues; on the other hand, SFN is only very weakly expressed in the normal alveolar epithelium. However, our immunohistochemical analysis using DAD autopsy specimens revealed that SFN was expressed mostly in the basal cells of the bronchioles and some alveolar type II cells that tended to show squamous metaplasia, which is generally observed in the mid to late phase of advanced DAD. Although SFN has been reported to be associated with lung adenocarcinoma[24, 25], no such association was observed in the serum samples of our lung cancer patients (Fig. 2 and data not shown). Instead, we would like to emphasize that the marked increase of serum SFN levels occurred specifically in patients with DAD.

The pathological changes of DAD proceed consistently through discrete but overlapping phases: the early exudative phase, the mid proliferative (organizing) phase, and the late fibrotic phase. They

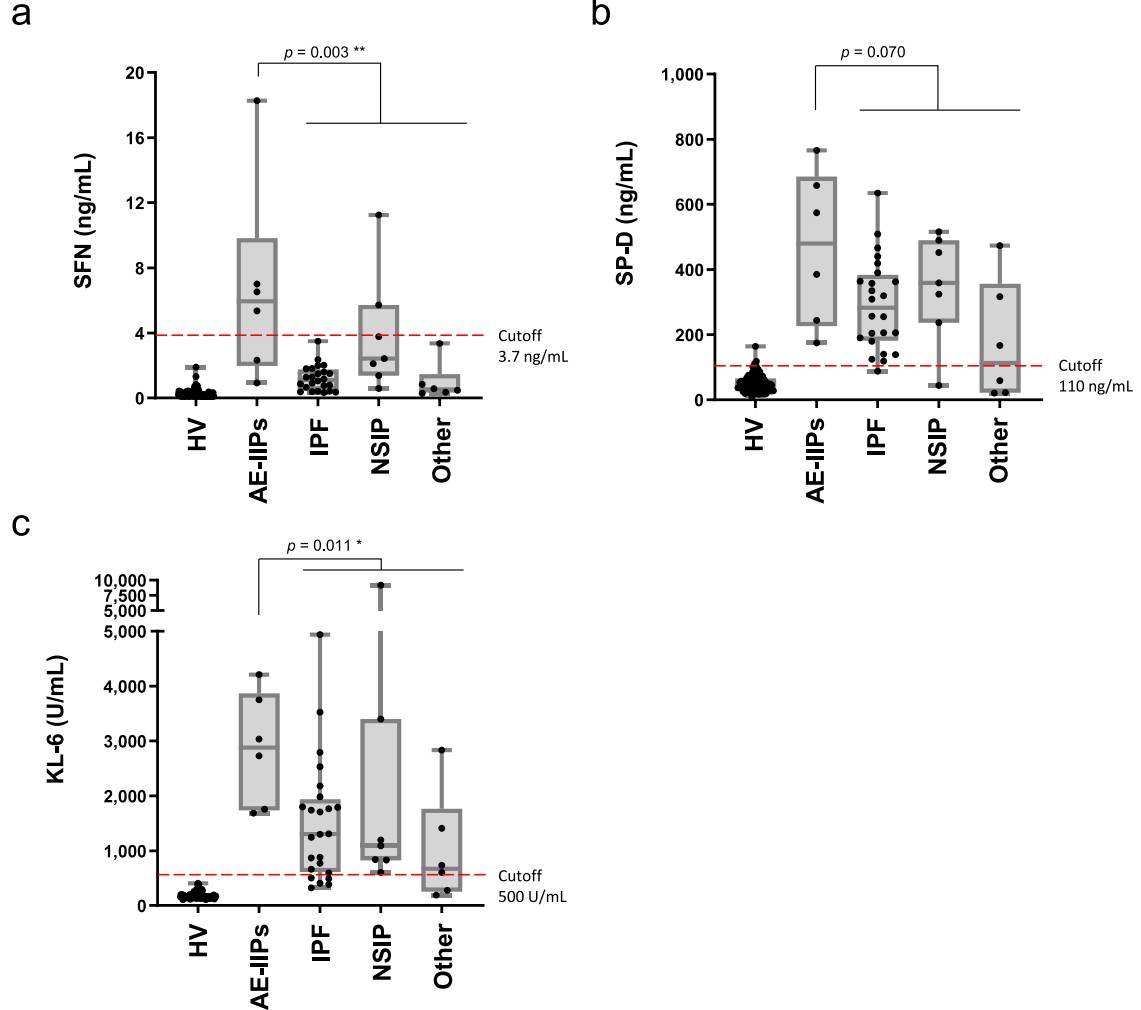

**Fig. 4 | Serum levels of SFN and known biomarkers in idiopathic interstitial pneumonias.** The distributions of SFN (**a**), SP-D (**b**), and KL-6 (**c**) are shown in box-plot graphs. Numbers of samples: HV ($n = 77$), AE-IIPs ($n = 6$), IPF ($n = 24$), NSIP ($n = 7$), Other ($n = 6$). The boxes indicate interquartile ranges (75% and 25%) and medians; whiskers show min and max values. Differences between the two groups were compared by a two-tailed Mann–Whitney $U$-test (unadjusted): *$p < 0.05$;

**$p < 0.01$. The broken lines indicate the respective cutoff values. HV healthy volunteer, AE-IIPs acute exacerbation of idiopathic interstitial pneumonias (DAD-type disease), IPF idiopathic pulmonary fibrosis, NSIP nonspecific interstitial pneumonia, Other other types of idiopathic interstitial pneumonia. Source data are provided as a Source data file.

reflect the global mechanisms of wound repair, and are thought to be involved in cell cycle regulation and apoptosis[26]. Denudation and apoptosis of alveolar epithelia may be important early features of acute lung injury. Guinee et al. reported that the expression of the transcription factor p53 as well as the expression of its transcription targets WAF1 and BAX were increased in type II pneumocytes in the lung of a patient with DAD[27, 28]. In addition, Bardales et al. showed that decreases in cell proliferation and increases in apoptosis were frequently observed in the lung tissues of patients with DAD, with both changes tending to be stronger in more severe cases, while apoptosis was undetectable in non-DAD patients[29]. Furthermore, Blázquez-Prieto et al. showed using an animal model that a p53 downstream factor, p21, is involved in the avoidance of apoptosis by lung epithelial cells, and suggested that p21-induced cellular senescence might be related to the repair of lung tissue[21]. This finding supports our data that the levels of released SFN, as well as those of the apoptosis indicator ccK18, were relatively low in the bleomycin- and Nutlin-3-treated A549 cells, which expressed a high level of p21 (Fig. 6d–f), although further research is needed.

It is also worth noting that SFN is a direct transcriptional target of p53[30]. In the current study, we showed that A549 and SAEC cells were

able to release SFN in vitro via apoptosis in the context of p53 activation (Figs. 6 and 7). These findings suggest that SFN may be upregulated by p53 activation when lung epithelial cells are damaged in patients with acute lung injury such as DAD, and extracellularly released via apoptosis, resulting in increased blood SFN levels. In particular, the release of SFN into the alveolar epithelium, which is the main field for gas exchange from the lungs to the blood, may contribute significantly to the increase in circulating SFN levels.

There have been many studies on the intracellular function and biological activity of SFN. The expression of SFN has been shown to be induced in a p53-dependent manner in response to DNA damage, and the progression of the G2/M phase of the cell cycle is arrested[30]. In addition, SFN is a marker of the differentiation of epidermal keratinocytes. An analysis based on skin biopsy samples revealed that the expression of SFN in keratinocytes in the squamous epithelium was stronger in areas close to the epidermis; moreover, an experiment using cultured cells found that the upregulation of SFN was closely related to a reduction in cell proliferation and the progression of cellular senescence[23].

Moreover, in addition to intracellular SFN, the role of extracellular released-SFN has also been reported. Previous studies have

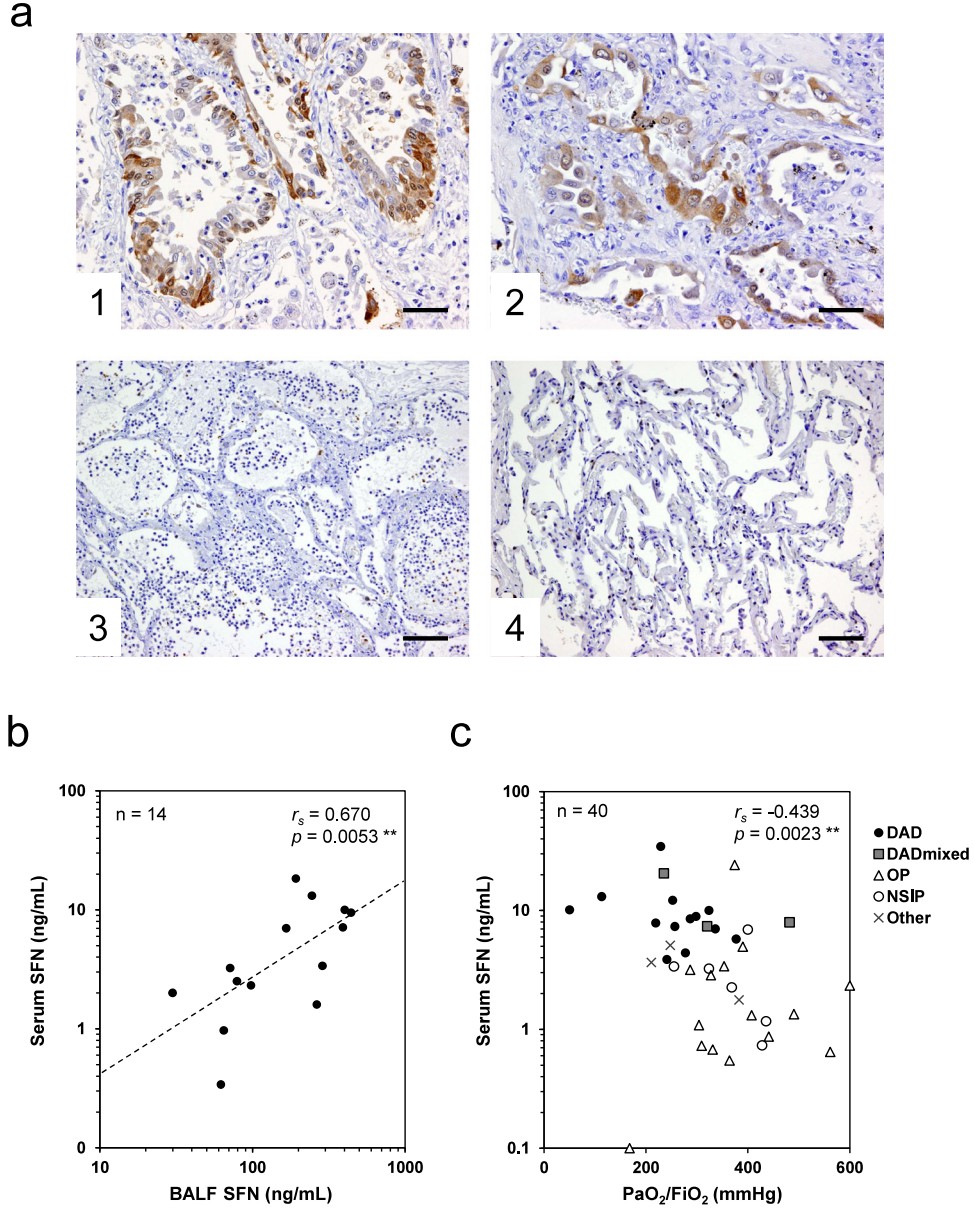

**Fig. 5 | SFN expression in lung tissue, and associations of serum SFN levels with those in BALF and with pulmonary functions. a** Representative immunohistochemical staining for SFN. Clear expression was observed in the basal cells of the bronchioles (**a1**, 200×) and alveolar type II cells (**a2**, 200×) in DAD specimens. Positive staining was not observed in the cells of alveolitis with bleeding in lung cancer patients (**a3**, 100×) and normal-looking lung tissue (**a4**, 100×). Scale bars are 50 μm (**a1**, **a2**) and 100 μm (**a3**, **a4**). **b** Correlation between serum SFN and BALF SFN. The data are for patients with DILD or IIPs ($n = 14$) from whom both serum and BALF samples were taken. BALF (ELF) SFN levels were estimated using the urea method. **c** Correlation between serum SFN levels and the ratio of arterial partial pressure of oxygen to fractional inspired oxygen ($PaO_2/FiO_2$) in the acute-phase DILD patients ($n = 40$). The values of the correlation coefficients $r_s$ and $p$ were given by Spearman's correlation analysis. Source data are provided as a Source data file.

demonstrated that the expression of SFN increases when keratinocytes are exposed to ultraviolet rays, and that keratinocyte-released SFN affects dermal fibroblasts via paracrine actions, facilitating the expression of matrix metalloproteinase (MMP)−1[31–33]. In addition, in keratinocytes, the extracellular expression of SFN protein is markedly increased compared to the intracellular expression[33], consistent with our data showing that the elevation of extracellularly released SFN after treatment with JNJ26854165 were much higher than that of the intracellular levels (Fig. 6d). The above-mentioned studies also suggested that keratinocyte-released SFN may be associated with skin wound healing and suppression of skin fibrosis[31–33]. Based on our results, we believe that the lung epithelia-released SFN may also be associated with pulmonary damage and remodeling. However, the

mechanistic link between the increase in SFN in blood and the pathophysiology of DAD requires further study.

DNA-damaging agents (DDAs) such as cisplatin, carboplatin, and bleomycin may potentially activate p53 in cells. However, in this study we observed no significant difference in the serum SFN levels between DILD patients who were administered DDAs and those who were administered other types of drugs (Supplementary Fig. 4). Importantly, these findings show that the measurement of serum SFN can detect DILD patients with DAD irrespective of the therapeutic agents used, including DDAs. On the other hand, high SFN levels were rarely observed in patients with lung cancer (or other types of cancer) (Table 3 and Supplementary Table 10). Therefore, when using serum SFN for monitoring the onset of DAD in oncology patients undergoing

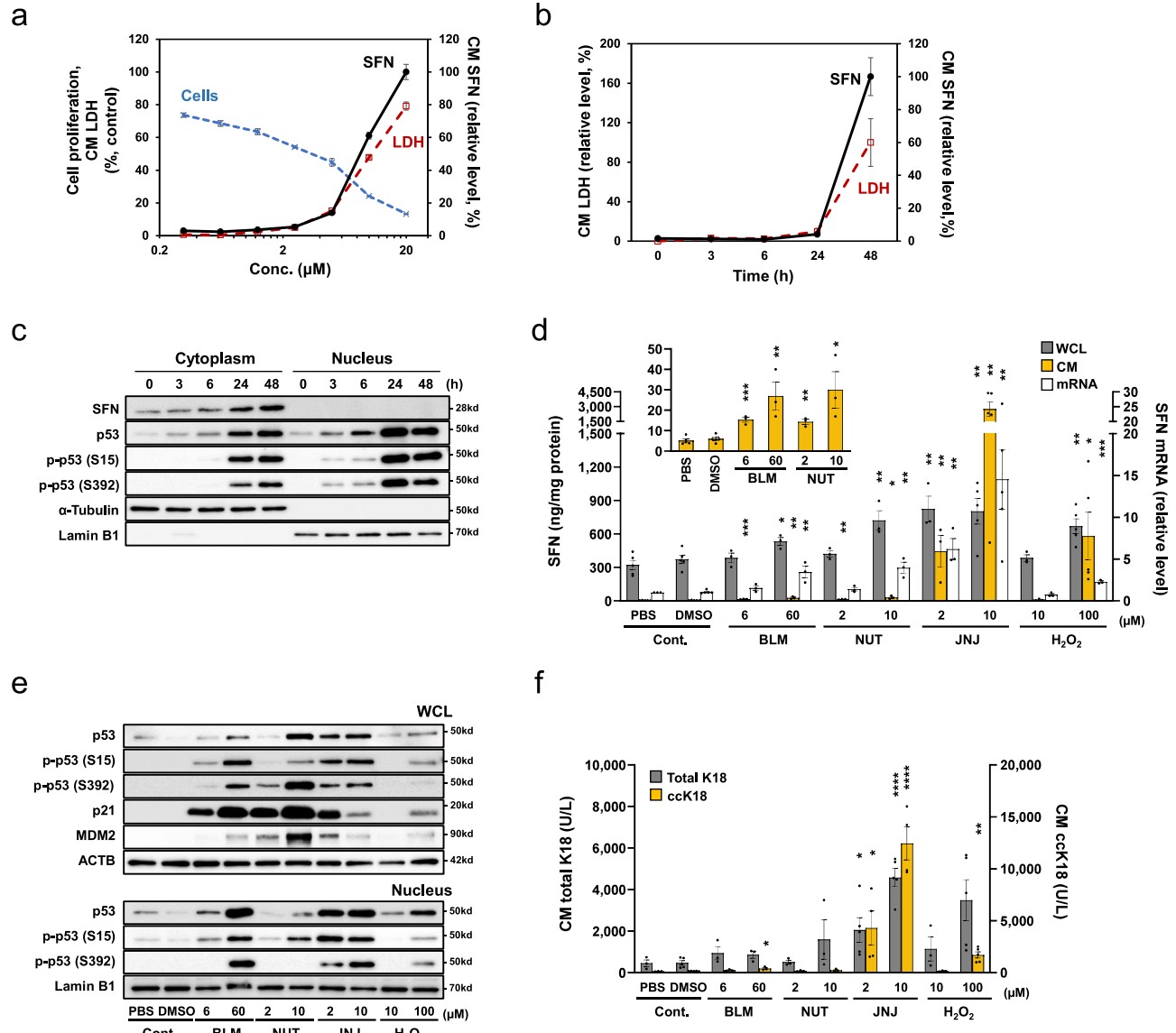

**Fig. 6 | Expression and extracellular release of SFN in vitro. a** Dose-dependent SFN release by JNJ26854165 treatment. Various concentrations of JNJ26854165 (JNJ) were added to the A549 cells. After 48 h, the cell proliferation and the levels of SFN and LDH released into the conditioned medium (CM) were measured. **b, c** Time-dependent SFN expression/release and nuclear translocation of p53 in response to JNJ. The A549 cells were treated with JNJ (10 μM). **b** At the indicated time points after the treatment, the CM was subjected to SFN ELISA and LDH assay. **c** Cells were collected and then fractionated into cytoplasmic and nuclear fractions, followed by Western blot analysis using the indicated antibodies. Anti α-tubulin and lamin B antibodies were used as a loading control of cytoplasmic and nuclear proteins, respectively. p-: phosphorylated at the amino acid number in parenthesis. **d–f** Comparison of the expression levels of SFN and p53-related proteins after treatment with various agents. The A549 cells were treated with the indicated

concentrations of bleomycin (BLM), nutlin-3 (NUT), JNJ and $H_2O_2$ for 48 h, and then SFN levels in CM and whole cell lysate (WCL) were measured by ELISA and the *SFN* mRNA levels were analyzed by qRT-PCR. **d** The amounts of SFN relative to the total amounts of proteins in WCL are shown. The CM SFN levels stimulated by BLM and NUT are enlarged and shown in the inset at the top left of the graph. The *SFN* mRNA levels were normalized with those of *ACTB* (β-actin). **e** Representative Western blot results for p53-related proteins in WCL and nucleus. **f** Keratin 18 (K18) and caspase cleaved K18 (ccK18) levels in CM. All graphs presented mean values ± SEM of $n = 3–5$ biologically independent experiments. Statistically significant differences by two-tailed Student's-test (unadjusted) in relation to the vehicle control are indicated with asterisks (*$p < 0.05$; **$p < 0.01$; ***$p < 0.001$; ****$p < 0.0001$). A vehicle control was used for each drug (DMSO for BLM, NUT, and JNJ, and PBS for $H_2O_2$). Source data are provided as a Source data file.

chemotherapy, it may also be necessary to measure the SFN levels prior to drug administration.

Several limitations of this study bear mentioned. First, although we were able to replicate the current findings, the number of patients enrolled was small and there were significant differences in demographic factors among them. However, this was somewhat inevitable due to the rarity of DAD-type DILD. Second, overall, as only the serum levels were examined, the findings of this study do not provide sufficient evidence to indicate a direct relationship between the pathology of DAD and SFN. Indeed, immunohistochemistry data were obtained

using autopsy samples from patients with DAD (not from patients assessed for serum SFN levels). Third, high levels of SFN were observed in certain DILD patients with the NSIP or OP pattern, as well as in some lung cancer patients. Therefore, to clarify the reason behind the high levels of SFN in patients without DAD, it is necessary to analyze a larger number of samples. Lastly, this was a retrospective study. Although we demonstrated that SFN increases in patients with DAD, irrespective of either disease types or suspected causative drug types, we cannot completely rule out a potential bias in the selection of patients. Currently, we are collecting serum samples chronologically and

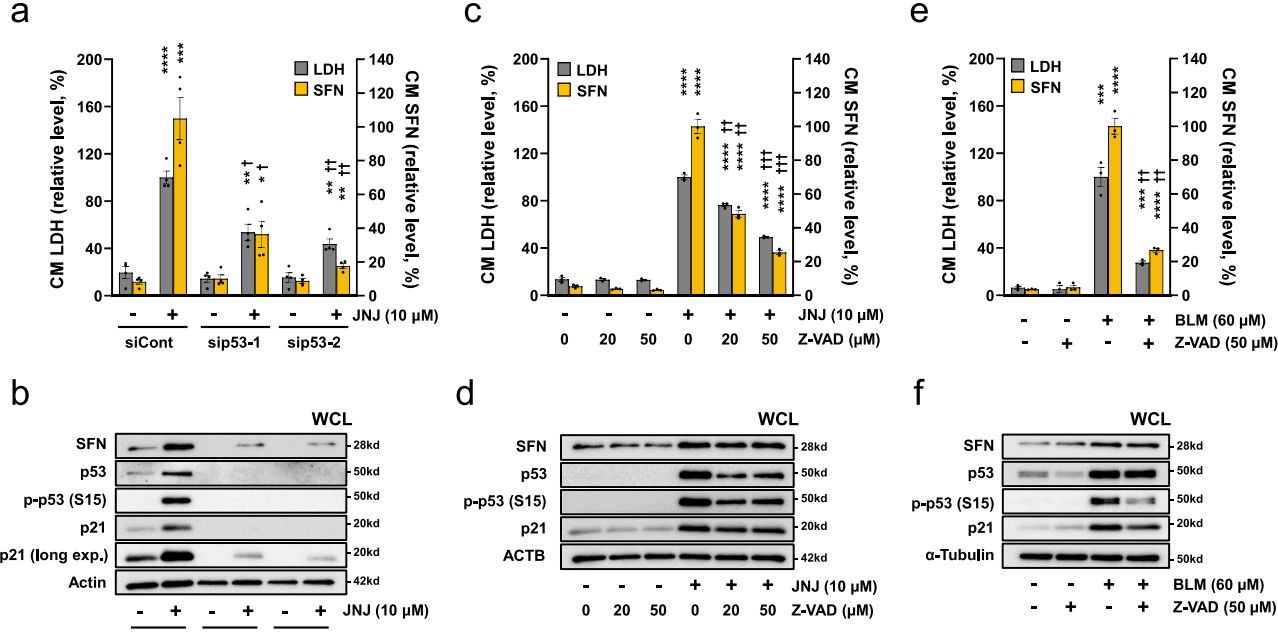

**Fig. 7 | Extracellular release of SFN via p53-dependent apoptosis. a, b** Effect of p53 knockdown on the release/expression of SFN. Twenty-four hours after transfection of A549 with control (siCont) or anti-p53 siRNAs (sip53-1, or −2), the cells were treated with JNJ (10 μM) for 48 h. The graph presented mean values ± SEM of $n = 4$ biologically independent experiments. **c, d** Effect of caspase inhibitor on release/expression of SFN. After treatment with the indicated concentrations of Z-VAD-FMK for 30 min, A549 cells were incubated with JNJ (10 μM) for 48 h. The graph presented mean values ± SEM of $n = 3$ biologically independent experiments. **e, f** The release/expression of SFN in the primary cultured small airway epithelial cells (SMECs). After treatment with or without Z-VAD-FMK (50 μM) for 30 min, SMEC were incubated with or without BLM (60 μM) for 48 h. The graph presented mean values ± SEM of $n = 3$ biologically independent experiments. The CM was subjected to SFN ELISA and LDH assay (**a, c, e**), and the WCL was subjected to Western blot analysis (**b, d, f**). To assess the knockdown efficiency of p53 by siRNA, the expression levels of p53, phosphor-p53 (S15) and p21 (a long exposure detection) were examined (**b**). Statistically significant differences by two-tailed Student's-test (unadjusted) in relation to the vehicle control are indicated with an asterisk (*$p < 0.05$; **$p < 0.01$; ***$p < 0.001$; ****$p < 0.0001$), and those in relation to the cells treated with JNJ (**a, c**) or BLM (**e**) are indicated with a dagger († $p < 0.01$; †† $p < 0.001$; ††† $p < 0.0001$). Source data are provided as a Source data file.

prospectively from patients prescribed EGFR-TKI and immune checkpoint inhibitors for future analysis. In addition, we are currently investigating the serum SFN expression mechanism using an animal model of acute lung injury. The analyses of these samples should shed light on the continuous changes in the levels of SFN that accompany the onset of DAD. Finally, the study has only been conducted in Japanese patients. Further research is necessary to verify whether the findings of this study are replicated in non-Japanese populations'.

In summary, in this study, we identified a protein biomarker for the diagnosis of DAD. SFN showed DAD-specific diagnostic performance, unlike known biomarkers such as KL-6 and SP-D. Since DILD and AE-IIP patients with DAD are likely to have a poor prognosis, it is necessary to make a diagnosis for such patients at the earliest period. Our data suggest that measuring the serum levels of SFN helps to distinguish DAD from other types of DILD, such as OP and NSIP, facilitating the diagnosis of DAD. Therefore, adopting this method may help in the determination of treatment plans for patients with DAD (in addition to the current diagnostic approach based on HRCT findings). Moreover, because SFN is also effective in discriminating the acute phase from the recovery phase, this protein may be useful for monitoring the response to treatment.

## Methods
### Clinical samples
Human Ethics approval was obtained for this study from each research ethics review committee of the participating institutes, Shinshu University, Nippon Medical School, Chiba University, Hiroshima University, the National Institute of Health Sciences, the Kihara Foundation, Astellas Pharma Inc., and Daiichi Sankyo Company. This study was performed in accordance with the Declaration of Helsinki and the Ethical Guidelines for Medical and Health Research Involving Human Subjects after approval by the National Institute of Health Sciences Research Ethics Committee (257-7). Serum and plasma samples were collected in the above first four hospitals with the approved protocols from patients who were suspected to have developed drug-induced interstitial pneumonia during the acute (around the most severe phase) and recovery phases of DILD, after informed consent was obtained.

DILD was diagnosed according to the Japanese diagnostic criteria by the respiratory specialists, as follows: (1) history of ingestion of a drug that is known to induce lung injury, (2) appearance of clinical manifestations after drug administration, (3) improvement of clinical manifestations after drug discontinuation, (4) exclusion of other causes of the clinical manifestations, and (5) exacerbation of clinical manifestations after resuming drug administration (challenge testing). Of note, criterion (5) was not used in this study due to ethical issues. Patient recovery was judged by respiratory specialists of each hospital at least two weeks following the onset of DILD, based on the recovery of clinical manifestations, improvement of lung imaging findings (e.g., HRCT), and improvement of oxygenation (e.g., SpO$_2$). Blood samples were collected from patients with any of the following conditions: lung cancer, non-tuberculosis mycobacteriosis, idiopathic interstitial pneumonia (IIP), lung disease associated with connective tissue disease, chronic obstructive pulmonary disease, bronchial asthma, and mycotic/bacterial pneumonia. Blood samples were also collected from tolerant control patients, most of whom were lung cancer patients undergoing chemotherapy without the development of DILD over the course of at least three months.

The following tubes were used for blood sampling: Vacutainer (Becton Dickinson, Franklin Lakes, NJ, USA) vacuum blood sampling tubes (6 mL) containing a blood coagulation accelerant for serum collection, and Vacutainer vacuum blood sampling tubes (7 mL)

containing EDTA-2Na for plasma collection. The tubes containing the blood were centrifuged for 10 min at $1300 \times g$, and serum and plasma were divided into small amounts and placed into polypropylene tubes with a screw cap. The samples were then stored in a deep freezer (−80 °C) until they were used.

Samples from healthy volunteers were collected at Yaesu Sakura-Dori Clinic after approval was obtained from the research ethics review committees of the participating organizations. The above-described protocols were followed to obtain serum and plasma samples. The inclusion criteria for healthy volunteers were as follows: (1) Japanese (self-declaration that all family members up to grandparents were Japanese) and diagnosed as healthy by their physician, (2) fasted for at least 14 h before blood collection (only drinking water was permitted), (3) no consumption of any drugs for at least 1 week, and (4) a normal body mass index ($18.5 \leq BMI < 25$). The exclusion criterion was (5) female individuals menstruating.

The "Discovery cohort" consisted of samples from DILD patients in the acute and recovery phases collected from April 2015 to November 2016 and samples from healthy volunteers. The Discovery cohort was used for biomarker discovery. Thereafter, an independent "Validation cohort" was created, consisting of samples from DILD patients collected from December 2016 to May 2019 and samples from the healthy volunteers that were not included in the Discovery cohort. Importantly, samples from patients with related lung diseases (lung cancer, IIP, lung disease associated with connective tissue disease, chronic obstructive pulmonary disease, nontuberculous mycobacteria, bronchial asthma, and bacterial and mycotic pneumonia) were included in the disease controls for biomarker evaluation in an exploratory analysis. In some cases, BALF samples collected at Chiba University were also used for analysis. DAD autopsy specimens from patients with interstitial pneumonia, and non-tumor autopsy or surgical specimens from patients with lung cancer, which were obtained at Shinshu University, were also used for immunohistochemical analysis.

## DILD disease classification
First, respiratory specialists who were extensively experienced in DILD diagnosis individually classified the pattern of histopathological subtypes of DILD at each hospital. However final diagnosis was confirmed by consensus in a meeting of specialists from the four sites. In this study, ~50% of the DILD patients underwent a transbronchial lung biopsy (TBLB) and/or bronchoalveolar lavage (BAL) testing; the remaining 50% did not undergo such invasive tests, due to the severity of their pneumonia symptoms. Finally, each DILD pattern was reviewed based on the patients' clinical findings (symptoms, medication history, laboratory test values, etc.) and HRCT imaging, but the specialists were blinded to the SFN values. The DAD pattern was diagnosed based on the presence of diffuse ground-glass opacities and/or infiltrated shadows in the bilateral lung fields on HRCT. Typical DAD patterns and DAD-dominant patterns (e.g., DAD > HP, DAD > OP) were classified as the DAD pattern. Cases with co-presence of the DAD and non-DAD patterns, but not DAD-dominant (e.g., OP > DAD, HP > DAD, DAD = HP) were classified as the DAD-mixed pattern. Patients with the DAD and DAD-mixed patterns were studied as the DAD-group. In contrast, patients with the OP pattern, NSIP pattern, and HP pattern were categorized as the non-DAD group.

## SOMAscan assay
The SOMAscan system (SomaLogic, Boulder, CO, USA) was used to perform proteomic analyses. Briefly, frozen plasma samples were sent to SomaLogic (Boulder, CO, USA) through the NEC Corp. (Tokyo, Japan). A total of 1310 proteins were obtained through the SOMAscan assay. For each protein probe, the relative fluorescence units (RFUs) were logarithmically transformed. Thereafter, a control group was set in which data from all recovered patients and healthy volunteers in the Discovery cohort were integrated. A search was performed for proteins whose abundance was significantly different between the DAD group (acute phase, with the DAD or the DAD-mixed patterns) and the control group, based on changes in mean values (fold change: FC), and effect size (Hedge's, $g$). All the SOMAscan proteomics data are shown in Supplementary Data 2.

## Measurement of SFN by in-house ELISA
Stratifin/14-3-3 (SFN) was measured using an in-house ELISA kit, which was developed using two commercially available anti-SFN mouse monoclonal antibodies, and recombinant human SFN protein produced in *E. coli* (NKMAX, Seongnam, Korea) as a standard. Briefly, 50 μL of a mouse anti-SFN monoclonal antibody (mAb) used as the primary capture antibody (clone CS112-2A8 [#05-632, Merck, Kenilworth, NJ, USA]) was added at a concentration of 4 μg/mL to Nunc MaxiSorp 96-well microtiter plates (Thermo Fischer Scientific, Waltham, MA, USA) and incubated at 25 °C for 18 h to prepare a solid phase. The plates were washed with a commercial washing buffer (Quantikine Wash Buffer; R&D Systems, Minneapolis, MN, USA) three times, and then incubated with the SuperBlock reagent (Thermo Fischer Scientific). The plates were then washed three times at 25 °C, and 50 μL samples (serum samples or standard SFN) diluted with General Serum Diluent (Immunochemistry Technologies, Bloomington, MN, USA) were added, followed by 50 μL of the assay buffer (SuperBlock reagent containing 2 M KCl and 50 μg/mL of the HBR-1 heterophilic antibody blocking reagent [Scantibodies Laboratory, Santee, CA, USA]), and the plates were stirred for 2 h. Following three washes, the SuperBlock reagent containing biotin-labeled secondary detection antibody (mouse anti-SFN mAb clone 3c3 [#WH0000286M1, Sigma-Aldrich, St. Louis, MO, USA, diluted 1:15,000]) was added and incubated for 1 h. After washing six times, plates were reacted with 10% SuperBlock/PBS solution containing Streptavidin-Poly HRP40 (Stereospecific Detection Technologies, Baesweiler, Germany). SFN was then quantitatively measured using the QuantaRed Enhanced Chemifluorescent HRP Substrate (Thermo Fischer Scientific).

This ELISA assay was validated with reference to the Japanese Guidelines on Bioanalytical Method Validation (Ligand Binding Assay) in Pharmaceutical Development (https://www.pmda.go.jp/files/000206208.pdf). We confirmed that the analytical parameters were sufficient for our current purposes within the criteria of the guidelines (Supplementary Table 2). Moreover, the ELISA method was used to measure SFN concentrations in the cell extracts, conditioned medium, and BALF. SFN concentrations in BALF were normalized with each concentration of urea in BALF and serum according to the method of Rennard et al.[34], and then represented as the concentration in epithelial lining fluid (ELF). The urea concentration was determined using a urea nitrogen colorimetric detection kit (Arbor Assays, Ann Arbor, MI, USA).

## Commercially available immunoassay kits
The serum levels of SP-D and KL-6 were measured using in vitro diagnostic kits, SP-D kit YAMASA EIA II (Yamasa, Chiba, Japan) and the E test TOSOH II (Tosoh, Tokyo), respectively. Serum kallistatin (KAL) was measured using DuoSet ELISA, and plasma C-C motif chemokine 18 (PARC) and plasma interleukin-1 receptor antagonist (IL-1Ra) were using Quantikine ELISA kits from R&D Systems (Minneapolis, MN, USA). Secreted phospholipase A2 (sPLA2) and apolipoprotein A-I (apo A-I) in plasma were measured using ELISA kits from Cayman Chemical (Ann Arbor, MI, USA) and Abcam (Cambridge, UK), respectively. All ELISA kits were used in accordance with the manufacturers' instructions. The K18 and ccK18 ELISA kits (VLV bio, Nacka, Sweden) were used for detecting apoptosis in vitro experiments.

## Immunohistochemistry of SFN
To investigate the expression and localization of SFN in the human lung, immunohistochemical analysis was performed with DAD autopsy

specimens from patients with DILD or IIPs, and non-tumor autopsy or surgical specimens from patients with lung cancer. The sections were deparaffinized, hydrated, and then immersed in 3% $H_2O_2$/methanol solution for 10 min at 25 °C for inactivation of endogenous peroxidase activity. After blocking nonspecific reactions with 10% normal goat serum, the sections were incubated with a primary antibody targeting SFN (diluted 1:1000; rabbit anti-SFN polyclonal antibody [HPA011105, Atlas Antibodies, Bromma, Sweden]) overnight at 4 °C. Visualization of antibody binding was performed using a VectaStain Elite ABC Kit (Vector Laboratories, Burlingame, CA, USA) and 3,3′-diaminobenzidine. All sections were counterstained with hematoxylin. The strength of staining was scored under a light microscope using the following criteria: −, almost all negative cells; ±, occasional slightly positive cells; +, some apparently positive lesions; ++, several positive lesions for SFN.

### In vitro experiments
The A549 cancer cell line derived from human type II pneumocytes was obtained from the JCRB Cell Bank (catalogue number JCRB0076) and cultured by the recommended methods. Briefly, the cells were plated at $5.0 \times 10^4$ cells per well of 12-well plates, or at $8.0 \times 10^5$ cells per 60 mm dish (for cell fractionation). Twenty-four hours later, bleomycin (Cayman Chemical, Ann Arbor, MI, USA), JNJ26854165 (Cayman Chemical), Nutlin-3a (Selleck Chemicals, Houston, TX, USA), or $H_2O_2$ solution (Nacalai Tesque, Kyoto, Japan) was added to the cells, and the cells were incubated for the indicated amount of time (Figs. 6 and 7). SFN in conditioned medium and cell extracts was measured using the in-house ELISA method or Western blot. The Viability/Cytotoxicity Multiplex Assay Kit (Dojindo Rockville MD, USA) was used to measure the cell proliferation and LDH activity. The p53 knockdown was performed with anti-p53 siRNAs (validated stealth siRNAs, ID nos. s607 and s605; Invitrogen) and a negative control (Silencer select negative control #1 siRNA; Invitrogen) using Lipofectamine RNAiMAX2000 (Invitrogen) according to the manufacturer's instructions. Z-Val-Ala-Asp-(OMe)-Fluoromethyl Ketone (Z-VAD-FMK, Cayman Chemical, Ann Arbor, MI, USA) was used as an apoptosis inhibitor.

A primary cultured small airway epithelial cells (originated from a 21-year-old Hispanic male [#KH-4209 (FC-0016), Kurabo, Osaka, Japan]), which were obtained with a fully informed consent, were cultured according to the manufacturer's recommendation, and used for replication of the findings observed in A549 cell line.

### Western blot analysis
Whole-cell lysates were prepared using RIPA buffer (Wako, Richmond, VA, USA). A Nuclear Extract Kit (Active Motif, Carlsbad, CA, USA) was used for the preparation of cytoplasmic and nuclear fractions from A549 cells according to the manufacturer's instructions. For detecting each protein, the following monoclonal or polyclonal antibodies were used: mouse anti-SFN mAb (clone CS112-2A8 [#05-632, Merck, USA; diluted 1:2000]), mouse anti-p53 mAb (clone DO-7 [#MA5-12557, Invitrogen, Carlsbad, CA, USA; 1:2000]), mouse anti-lamin b1 mAb (clone 3C10G12 [#66095-1-Ig, Proteintech, USA; 1:10,000]), rabbit anti-phospho-p53 Ser15 pAb (#9284, Cell Signaling Technology, Leiden, the Netherlands; 1:2000), rabbit anti-phospho-p53 Ser392 pAb (#9281, Cell Signaling Technology; 1:2000), rabbit anti-MDM2 mAb (clone D1V2Z [#86934, Cell Signaling Technology; 1:2000]), mouse anti-α-tubulin mAb (clone DM1A [#3873, Cell Signaling Technology; 1:10,000]), rabbit anti-p21 Waf1/Cip1 mAb (clone 12D1 [#2947, Cell Signaling Technology; 1:2000]) and β-actin rabbit pAb (20536-1-A, Proteintech; 1:10,000). The HRP-conjugated horse anti-mouse IgG (#7076, Cell Signaling Technology; 1:10,000), and goat anti-rabbit IgG (#7074, Cell Signaling Technology; 1:10,000) were used as the secondary antibodies. Chemiluminescence was detected using an ImageQuant LAS 4000 mini imaging system (GE Healthcare, Uppsala, Sweden) with the ECL select HRP substrate reagent (GE Healthcare).

### qRT-PCR analysis
*SFN* mRNA was measured by real-time PCR. Total RNA was extracted from the cells using RNeasy spin columns (Qiagen, Hilden, Germany). The extracted total RNA was then quantitatively measured using a One Step SYBR PrimeScript RT-PCR Kit (Takara, Kyoto, Japan) and the ABI PRISM 7700 PCR thermal cycler. The β-actin gene (*ACTB*) was used as the internal standard. The PCR primers with the following sequences were used: SFN-F, 5′-TGACGACAAGAAGCGCATCAT; SFN-R, 5′-GTAG TGGAAGACGGAAAAGTTCA; ACTB-F 5′-CACCATTGGCAATGAGC GGTTC; ACTB-R, 5′-AGGTCTTTGCGGATGTCCACGT.

### Statistic and reproducibility
All experiments for quantitative analysis and representative images were reproduced with similar results for at least three times. In group comparisons of demographics and clinical parameters, the Kruskal Wallis test was used for continuous variables, and the Fisher's exact test was used for categorical variables, as appropriate. If a significant difference was found by the analysis of variance, Dunn's multiple comparison test was applied to evaluate differences from the DAD group. To find the minor differences that can influence the analysis, correction for multiple comparisons was not applied in this study. The difference in the degree of changes in the amounts of proteins, as per the SOMAscan analysis, was evaluated via the calculation of Hedge's $g$ values. Statistically significant differences between two groups of candidate marker proteins were calculated by the Mann–Whitney $U$ test. The diagnostic performance of each candidate was evaluated based on the AUC values of ROC curves. The cut-off value for SFN was set based on Youden's index in comparison with the DAD group and the tolerant control. Spearman's nonparametric analysis was used to evaluate the correlation between variables. Unadjusted two-tailed $p < 0.05$ was considered statistically significant. All data were analyzed using the following software: GraphPad PRISM (version 8.4.3; Graph-Pad Software, San Diego, CA, USA) and Microsoft Excel.

### Reporting summary
Further information on research design is available in the Nature Research Reporting Summary linked to this article.

## Data availability
The data that support the findings of this study including clinical information, and SOMAscan and ELISA data are available within the paper and its Supplementary Information. The source data underlying all figures and supplementary figures are provided as a Source Data file. The immunohistochemistry images for SFN expression in normal tissues referenced in this study are available in the Human Protein Atlas database (https://www.proteinatlas.org/ENSG00000175793-SFN/tissue). Source data are provided with this paper.

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

## Acknowledgements

We thank Kazuishi Kubota, Naoya Wada, and Kazumi Ito for their technical advice. Financial support was received in the form of grants from the Japan Agency for Medical Research and Development (nos. 19mk01045 and 20mk01073j to N.A., A.G., K.T., N.H., K.S., Y.O., Yoshiro S., and M.H.), and a KAKENHI Grant-in-Aid for Scientific Research (C) from the Ministry of Education, Culture, Sports, Science and Technology of Japan (no. g20K07366 to N.A.).

## Author contributions

Conceptualization: N.A., Yoshiro S., and M.H. Data curation: K.S. and Yoshiro S. Methodology: N.A., R.N., M.S., K.T., Kazuhiko M., T.N., T.T., K.O., Yoshiro S., and M.H. Investigation: N.A., S.M., Keiko M., R.N., and T.T. Funding acquisition: N.A., K.S., A.G., K.T., N.H., Y.O., Yoshiro S., and M.H. Project administration: T.I., Yoshiro S., M.S., K.T., Kazuhiko M., T.N., and Y.O. Resources: A.U., M.A., Yoshinobu S., T.K., Y.H., A.G., K.T., N.H., and M.H. Supervision: Y.O., Yoshiro S., and M.H. Writing—original draft: N.A., K.O., and Yoshiro S. Writing—review & editing: All authors.

## Competing interests

N.A., R.N., M.S., T.N., Yoshiro S., and M.H. are inventors of patents related to biomarker proteins involved in the diagnosis of DILD as filed by the National Institute of Health Science (PCT/JP2020/28866). The remaining authors declare no competing interests.
