## [Peer Review File · Nature Communications]

Stratifin as a Novel Diagnostic Biomarker in Serum for Diffuse Alveolar DamageREVIEWER COMMENTS

Reviewer #1 (Remarks to the Author):

Thank you for asking me to review this manuscript by Arakawa et al titled "Stratifin as a Novel Diagnostic Biomarker in Serum for 2 Diffuse Alveolar Damage". The authors report the role of serum Stratifin in identifying individuals with drug induced diffuse alveolar damage from those with other forms of drug induced ILD. Their data suggest that Stratifin identifies DI-DAD patients from healthy controls with a high degrees of accuracy and from other groups of patients with ILD of any cause with moderate accuracy. Although Stratifin has been reported as a factor associated with lung cancer progression it has not previously been reported as a biomarker of either diffuse alveolar damage or interstitial lung disease. I have a number of criticisms of the manuscript as presented.

1) It is not entirely clear from the methods how a diagnosis was assigned to individual subjects. Given that the clinically assigned diagnosis acts as the gold standard for the biomarker comparisons it is important that this is explained in absolute detail. Were all diagnoses assigned by central MDT? If so, how did the central diagnosis differ from local diagnosis? What information was used to make the diagnosis – baseline imaging and history alone, or all longitudinal data or were biopsies or autopsies available for some subjects? Were there outcome differences between groups?

2) It seems likely that there were differences in disease severity (hypoxia, oxygen requirements etc etc) between the DAD and the non-DAD groups. Did the authors explore the possibility that these differences contributed to some of the biomarker differences seen between groups? At the very least, it would be of relevance to know whether biomarker levels predicted outcomes such as length of hospital stay or mortality.

3) The data suggest that Stratifin is probably a marker of diffuse lung injury. It would be of interest to see if other forms of DAD (i.e. covid related, post-surgical etc) show elevated levels of Stratifin. Similarly, it would be of interest to compare performance of Stratifin to other markers associated with DAD such as sRAGE.

4) A number of the comparisons described in the manuscript come from analysis of the combined discovery and validation cohorts. These analyses need to be sign posted as exploratory.

5) The abstract suggests that there were 432 patients assessed in the analyses. Whilst this may be true it disguises the fact that the analyses of meaning involved 10 and 16 patients with DAD in the discovery and validation cohorts and 30 and 28 patients in the non-DAD drug induced ILD cohorts. Whilst these small numbers do not invalidate the reported findings they do suggest that some caution should be applied to over interpreting findings.

6) The authors derive their Youden index in the combined cohort. This should really be calculated in the discovery cohort and confirmed in the validation cohort.

7) A549 cells are not really representative of alveolar epithelial cells. They are a cell line derived from non-small cell cancer. The A549 experiments presented are not really relevant for a biomarker paper. However, if they are to remain, I would suggest replicating the findings in small airway epithelial cells (such as are available from Lonza and other suppliers).

8) The study has only been conducted in Japanese patients (in fact controls were only permitted if they could show three generations of Japanese relatives). As such, it is hard to know how well the findings from this research may translate outside of Japan and this should be mentioned in the limitations.

Reviewer #2 (Remarks to the Author):

The authors use the A549 cancer cells to assess whether lung epithelial cells can release SFN. This is not the correct model in vitro model to assess their hypothesis. They should have used primary cells derived from surgical specimens of human lungs. The lines 295-297 deliver a superficial statement, Nutlin is a P53 inducer and the statement "bleomycin and hydrogen peroxide, both associated with ARDS and bleomycin-induced pneumonia" is not accurate (especially to the audience familiar with ARDS). In figure 6D it appears that H₂O₂ induces p21 in high concentrations, but does not affect p53 expression levels. Please explain, and take into consideration a plethora of studies

suggesting otherwise (e.g. PMID: 27375834). Furthermore, the authors should proceed with time- and dose- dependent treatments of BLM, NUT, JNJ and H₂O₂ in the cells, and evaluate the corresponding effects in the nuclear translocation of P53. They should also evaluate the expression of total and 2) phospho MDM2, MDM4, and p53 (e.g. Ser 15, Ser 392) in the corresponding experiments. Those post translational modifications modulate p53 expression and activation. All those experiments should be repeated in cells that express more or less p53 due to transfections and transductions. Indeed, the authors should take into consideration that P53 possess multifarious roles in lung physiology. For example please see the study by Albaiceta'sgroup (<https://doi.org/10.1016/j.trsl.2021.01.008>)

Reviewer #3 (Remarks to the Author):

In this manuscript, SFN is elevated in DILD patients with DAD and as a biomarker. The statistical methods are validated. However, I have the following comments:

1. As a biomarker, the relationships between SFN and patient demographic variables, such as age, gender, race and others, should be analysed.
2. among different groups (DAD, DILD, and others), are the patient demographic/clinical variables significantly different? Some statistical tests and the corresponding p-values should be provided.
3. Using a logistic model, a panel of biomarkers may be better.

Detailed responses to the reviewers' comments:

Reviewer #1:

1) It is not entirely clear from the methods how a diagnosis was assigned to individual subjects. Given that the clinically assigned diagnosis acts as the gold standard for the biomarker comparisons it is important that this is explained in absolute detail. Were all diagnoses assigned by central MDT? If so, how did the central diagnosis differ from local diagnosis? What information was used to make the diagnosis – baseline imaging and history alone, or all longitudinal data or were biopsies or autopsies available for some subjects? Were there outcome differences between groups?

Response: Thank you very much for your important remarks. Firstly, respiratory specialists who were extensively experienced in DILD diagnosis individually classified the histopathological subtypes of DILD at each hospital. However final diagnosis was confirmed by consensus in a meeting of specialists from the four sites. All cases were reviewed based on the clinical findings (symptom, medication history, laboratory test values, etc.) and HRCT imaging, although biopsy data (TBLB) were also used to aid in diagnosis in some cases. We added more detailed information in the Methods section [**revised text in red, p.25**]. We also described about the differences in outcomes between the DAD and non-DAD groups [**revised text in red, p.13, ll. 269-271**].

2) It seems likely that there were differences in disease severity (hypoxia, oxygen requirements etc etc) between the DAD and the non-DAD groups. Did the authors explore the possibility that these differences contributed to some of the biomarker differences seen between groups? At the very least, it would be of relevance to know whether biomarker levels predicted outcomes such as length of hospital stay or mortality.

Response: Thank you for this important suggestion. We added the data and the corresponding explanation that there was no significant difference in SFN values between patients who recovered from DAD-type DILD and those who did not, and no correlation between SFN values and the number of days required for DILD treatment [**revised Supp. Fig. 5, revised text in red, p.13, ll.268-276**].

3) The data suggest that Stratifin is probably a marker of diffuse lung injury. It

would be of interest to see if other forms of DAD (i.e. covid related, post-surgical etc) show elevated levels of Stratifin. Similarly, it would be of interest to compare performance of Stratifin to other markers associated with DAD such as sRAGE.

Response: Thank you for the important point. A study of serum levels of stratifin in COVID-19 patients is now on going, and will be reported elsewhere. In addition, we hope to perform a comparison analysis of sRAGE and SFN as another study in the future.

4) A number of the comparisons described in the manuscript come from analysis of the combined discovery and validation cohorts. These analyses need to be sign posted as exploratory.

Response: As the reviewer suggested, we added explanations where applicable to the exploratory analysis [revised text in red, pp.5-6, ll.93-96; p.9, ll.175-177]

5) The abstract suggests that there were 432 patients assessed in the analyses. Whilst this may be true it disguises the fact that the analyses of meaning involved 10 and 16 patients with DAD in the discovery and validation cohorts and 30 and 28 patients in the non-DAD drug induced ILD cohorts. Whilst these small numbers do not invalidate the reported findings they do suggest that some caution should be applied to over interpreting findings.

Response: Thank you very much for indicating this important point. We revised the text in the abstract as follows: The study included two independent cohorts (including totally 26 patients with DAD) and controls (total 432 samples). [revised text in red, p.3].

6) The authors derive their Youden index in the combined cohort. This should really be calculated in the discovery cohort and confirmed in the validation cohort.

Response: In general, the reviewer's point is correct. However, we calculated the Youden index-based cutoff value, 3.6 ng/mL, from the results of a combined cohort with a larger number of data, because of the small sample size in the Discovery cohort. In addition, we confirmed that the cutoff values yielded when analyzed separately for the Discovery and Validation cohort were also in the very close range of 3.6 to 3.7 ng/mL. Therefore, we determined that our cutoff value of 3.6 ng/mL was reasonable. [revised text in

red, p.11, ll.218-220]

7) A549 cells are not really representative of alveolar epithelial cells. They are a cell line derived from non-small cell cancer. The A549 experiments presented are not really relevant for a biomarker paper. However, if they are to remain, I would suggest replicating the findings in small airway epithelial cells (such as are available from Lonza and other suppliers).

Response: Thank you for this important suggestion. Reviewer#2 also mentioned this point. We examined replicating the finding observed in A549 cell line, using a primary cultured small airway epithelial cells (SAECs, Kurabo, Osaka, Japan). The upregulation of SFN expression/release with p53 activation and its attenuation by Z-VAD-FMK were also observed in the SAECs in addition to findings using the A549 cell line [**revised. Fig. 6i and l, revised text in red, p.15 ll.326-327, p17, ll.363-367, p30, ll.676-678**].

8) The study has only been conducted in Japanese patients (in fact controls were only permitted if they could show three generations of Japanese relatives). As such, it is hard to know how well the findings from this research may translate outside of Japan and this should be mentioned in the limitations.

Response: According the suggestion, we added the following explanation into the limitations: the study has only been conducted in Japanese patients. Further research is necessary to verify whether the findings of this study are replicated in non-Japanese populations'. [**revised text in red, p22, ll.484-486**].

Broadly speaking this is an interesting study in that it is the first to identify Stratifin as a biomarker of diffuse alveolar damage. However, this is a small retrospective study and the comparator groups are opportunistic rather than those which might be chosen if designing a prospective study. Ultimately, it merits publication after revision but I am not sure that it is of the level of priority to merit publication in Nature Comms

Response: We agree with the reviewer about the importance of a prospective study with a large number of patients to assess the biomarker utility. However, unlike a biomarker study for common diseases, prospective collection of DILD cases is quite difficult due to the rare incidence of this adverse drug reaction (even at our four large collaborative hospitals), making

it difficult to perform the prospective assessment of biomarkers. Indeed, it took over 5 years to collect 84 DILD cases including 26 patients with DAD.

In the current cohort, the 95% confidence intervals of the diagnostic performance for discriminating DAD vs non-DAD (AUC of the ROC) of SFN were 0.83–0.97 (revised Fig. 3b). This data showed that SFN has an advantage not found in the existing biomarkers, SP-D (0.48–0.73) and KL6 (0.44–0.69). These results were shown in two independent cohorts, and thus validated. We believe that these results are sufficient to suggest the usefulness of SFN as a DAD-type DILD marker.

Reviewer #2:

The authors use the A549 cancer cells to assess whether lung epithelial cells can release SFN. This is not the correct model in vitro model to assess their hypothesis. They should have used primary cells derived from surgical specimens of human lungs. The lines 295-297 deliver a superficial statement, Nutlin is a P53 inducer and the statement "bleomycin and hydrogen peroxide, both associated with ARDS and bleomycin-induced pneumonia" is not accurate (especially to the audience familiar with ARDS). In figure 6D it appears that H2O2 induces p21 in high concentrations, but does not affect p53 expression levels. Please explain, and take into consideration a plethora of studies suggesting otherwise (e.g. PMID: 27375834). Furthermore, the authors should proceed with time- and dose-dependent treatments of BLM, NUT, JNJ and H2O2 in the cells, and evaluate the corresponding effects in the nuclear translocation of P53. They should also evaluate the expression of total and 2) phospho MDM2, MDM4, and p53 (e.g. Ser 15, Ser 392) in the corresponding experiments. Those post translational modifications modulate p53 expression and activation. All those experiments should be repeated in cells that express more or less p53 due to transfections and transductions. Indeed, the authors should take into consideration that P53 possess multifarious roles in lung physiology. For example please see the study by Albaiceta'sgroup (<https://doi.org/10.1016/j.trsl.2021.01.008>)

–the A549 cell line is not the correct model

Response: We agree with the reviewer about the importance of investigation using primary-cultured normal human lung epithelial cells, but not A549 cell line. In addition, Reviewer#1 also mentioned this point. According to this comment, by using a primary cultured small airway epithelial cells (SAECs), we examined replicating the finding observed in A549 cell line. The upregulation of SFN expression/release with p53 activation and its attenuation by an apoptosis inhibitor Z-VAD-FMK were also observed in the SAECs as well as the A549 cell line [**revised. Fig. 6i and l, revised text in red, p.15 ll.326-327; pl7, ll.363-367; p30, ll.676-678**].

–The statement "bleomycin and hydrogen peroxide, both associated with ARDS and bleomycin-induced pneumonia" is not accurate.

Response: Thank you very much for indicating this important point. We revised the text accordingly [**revised text in red, p.15, ll. 327-333**].

–In figure 6D it appears that H₂O₂ induces p21 in high concentrations, but does not affect p53 expression levels.

Response: As the reviewer points out, the p53 upregulation appears to be weak in the H₂O₂ (100 μM)-treated cells in the whole cell lysate (original version of Fig. 6D). Our Western blot analysis reproduced this result [**revised Fig. 6e**]. According to the reviewer's suggestion, we performed further analysis to investigate the time course of p53 levels in cytoplasm and nucleus by H₂O₂. As a result, we found that the time course of p53 accumulation following H₂O₂ treatment was different from that by JNJ26854165 treatment. In the case of JNJ26854165 (10 μM) treatment, p53 accumulation in the cytoplasm was more pronounced at 3 hours after exposure to this agent, with simultaneous elevation in the nuclei for total p53 (from 3 hours after the treatment) and phosphorylated-p53 (Ser15 and Ser392, 24 hours after the treatment), and the effect lasted up to 48 hours [**revised Fig. 6c**]. In contrast, H₂O₂ (100 μM) caused the accumulation of p53 at 3 hours after the exposure, but the effect had almost disappeared after 24 hours, resulting in a reduced level of p53 after 48 hours (see the figure below). Of note, the p21 expression level was persistently increased up to 48 hours. Therefore, we observed an apparent discrepancy between the expression levels of p21 and p53 in the whole cell lysate [**revised Fig. 6e**]. However, we believe that this data (figure below) was not necessary for the underlying story in our revised paper, so we did not include it in the test/figure.

Fig. 1 Time course of expression of SFN and p21, and nuclear translocation of p53.

–the authors should proceed with time- and dose- dependent treatments of BLM, NUT, JNJ and H2O2 in the cells, and evaluate the corresponding effects in the nuclear translocation of P53. They should also evaluate the expression of total and 2) phospho MDM2, MDM4, and p53 (e.g. Ser 15, Ser 392) in the corresponding experiments. All those experiments should be repeated in cells that express more or less p53 due to transfections and transductions. Indeed, the authors should take into consideration that P53 possess multifarious roles in lung physiology. For example please see the study by Albaiceta'sgroup (<https://doi.org/10.1016/j.trsl.2021.01.008>)

Response: As mentioned above, we performed further analysis to investigate the time course of accumulation and nuclear translocation of p53 following JNJ26854165 and H₂O₂ treatment. We observed that the time course of intracellular SFN expression was correlated with the accumulation of p53 and the level of phosphorylated p53 (Ser15 and Ser392) in the nucleus by JNJ26854165 [**revised Fig. 6c**]. The nuclear p53 levels were also increased in the bleomycin- and nutlin-3-treated cells in which the released level of SFN were slight [**revised Fig. 6d and 6e**], suggesting that p53 inhibitory factors such as MDM2 or MDM4 are unlikely to be involved in the attenuation of SFN release in these cells. Indeed, the level of MDM2 was not correlated with the SFN release [**revised Fig. 6d and 6e**]. On the other hand, we noticed that the expression levels of p21 were markedly higher in these slight SFN-releasing cells (Bleomycin- and Nutlin-3-treated cells) than in the high level SFN-releasing cells (JNJ26854165 and H₂O₂-treated cells) [**revised Fig. 6e, upper panel**]. This data suggests that the anti-apoptotic effect

of p21, as described by Albaiceta's group, was strong in the slight SFN-releasing cells. This notion is supported by the results showing that the release of SFN was linked with the release of an apoptosis marker ccK18 [revised Fig. 6f].

In addition, according to the reviewer's suggestion, we further investigated the effect of anti-p53 siRNA transfection on the intracellular/extracellular expression of SFN. As a result, we found that siRNA-mediated p53 knockdown suppressed the extracellular release of SFN (and LDH) by JNJ26854165 [revised Fig. 6g and j]. Moreover, the increased level of extracellularly released SFN in the JNJ26854165-treated cells was significantly reduced by treatment with an apoptosis inhibitor, Z-VAD-FMK [revised Fig. 6h and 6k]. We finally analyzed a primary cultured SAECs for replicating these findings obtained in A549 [revised Fig. 6i and 6l]. We believe these results will help to elucidate the mechanism of SFN release, as you kindly suggested; *ie*, the extracellular release of SFN was suggested to occur via p53-dependent apoptosis. We have included a description and discussion of these results in the revised version of the manuscript [pp. 15-17, 19, 29-30]. We also included the study of Albaiceta's group (<https://doi.org/10.1016/j.trsl.2021.01.008>) as **reference No. 29**. We thank the reviewer for the important tips in regard to this analysis and the interpretation of results.

Reviewer #3:

In this manuscript, SFN is elevated in DILD patients with DAD and as a biomarker. The statistical methods are validated. However, I have the following comments:

1. As a biomarker, the relationships between SFN and patient demographic variables, such as age, gender, race and others, should be analysed.

Response: Thank you for this important suggestion. We added the data and the corresponding explanation about correlations between the serum SFN values and the age and sex of patients [revised Supp. Fig. 3, revised text in red, pp.11-12, ll.235-239].

2. among different groups (DAD, DILD, and others), are the patient demographic/clinical variables significantly different? Some statistical tests and the corresponding p-values should be provided.

Response: Thank you for this important point as well. We added the data showing the differences among patient groups as suggested. These include:

- Differences in age, sex, and the sample number in the acute/recovery phase [revised Table 1, revised text in red, p.6, ll.111-116; p.21; p.31, ll.692-697].

- Differences in clinical chemistry and respiratory parameters [revised Supp. Table 1, revised text in red, pp.6-7, ll.116-121; p.31, ll.692-697]

3. Using a logistic model, a panel of biomarkers may be better.

Response: As suggested, we performed further analyses to determine whether known biomarkers (SP-D, KL-6 and CRP) and new biomarker candidates identified in this study—i.e., kallistatin (KAL) and apo-AI, which showed relatively good performance in discriminating the DAD group from the non-DAD or recovery groups [revised Supp. Fig. 1] would improve the diagnostic performance of SFN. For this purpose, we performed a ROC analysis using the logistic regression model with JMP software (ver. 16; SAS, Cary, NC).

As shown in Table 1 below, no biomarker combinations were found to significantly outperform SFN alone for discrimination between the DAD group and non-DAD group or between the DAD group and Recovery group.

Table 1. Comparison the diagnostic performance of combined biomarker models.

Model	DAD group vs non-DAD group		DAD group vs Recovery	
	AUC (95% CI)	Sensitivity at 85% specificity	AUC (95% CI)	Sensitivity at 85% specificity
SFN alone	0.90 (0.83 - 0.97)	80.8	0.93 (0.88 - 0.99)	92.3
SFN + SP-D	0.91 (0.84 - 0.98)	80.0	0.88 (0.80 - 0.96)	72.0
SFN + KL-6	0.90 (0.83 - 0.97)	80.8	0.85 (0.74 - 0.96)	80.8
SFN + CRP	0.86 (0.78 - 0.94)	69.2	0.89 (0.82 - 0.96)	84.6
SFN + KAL	0.88 (0.80 - 0.96)	78.3	0.92 (0.86 - 0.98)	73.9
SFN + Apo-A1	0.85 (0.77 - 0.94)	76.9	0.89 (0.82 - 0.97)	76.9

Data of the Combined cohort are shown.

Until now, there have been no known blood biomarkers for DAD. In addition, to the best of our knowledge, no study has reported the detailed behavior of SFN released into blood. We believe that the main finding of this paper is thus the discovery and validation of SFN as a potential serum biomarker for DAD, which we consider a novel finding. Because we demonstrated the mechanisms of SFN release in detail [revised Fig. 6], if possible, we would like this report to remain focused on the usefulness of SFN alone.

REVIEWER COMMENTS

Reviewer #1 (Remarks to the Author):

The authors have adequately addressed my prior comments. The addition of SAEC experiments enhances the pre-clinical work. The biomarker analyses are limited by small numbers but this is addressed by the authors in their discussion.

Reviewer #2 (Remarks to the Author):

The authors have addressed my comments.

Reviewer #3 (Remarks to the Author):

The manuscript has been greatly improved.